# The small molecule ISRIB rescues the stability and activity of Vanishing White Matter Disease eIF2B mutant complexes

Yao Liang Wong[1†]*, Lauren LeBon[1†], Rohinton Edalji[2], Hock Ben Lim[2], Chaohong Sun[2], Carmela Sidrauski[1]*

[1]Calico Life Sciences LLC, South San Francisco, United States; [2]Discovery, Global Pharmaceutical Research and Development, AbbVie, North Chicago, United States

**Abstract** eIF2B is a dedicated guanine nucleotide exchange factor for eIF2, the GTPase that is essential to initiate mRNA translation. The integrated stress response (ISR) signaling pathway inhibits eIF2B activity, attenuates global protein synthesis and upregulates a set of stress-response proteins. Partial loss-of-function mutations in eIF2B cause a neurodegenerative disorder called Vanishing White Matter Disease (VWMD). Previously, we showed that the small molecule ISRIB is a specific activator of eIF2B (*Sidrauski et al., 2015*). Here, we report that various VWMD mutations destabilize the decameric eIF2B holoenzyme and impair its enzymatic activity. ISRIB stabilizes VWMD mutant eIF2B in the decameric form and restores the residual catalytic activity to wild-type levels. Moreover, ISRIB blocks activation of the ISR in cells carrying these mutations. As such, ISRIB promises to be an invaluable tool in proof-of-concept studies aiming to ameliorate defects resulting from inappropriate or pathological activation of the ISR.
DOI: https://doi.org/10.7554/eLife.32733.001

*For correspondence:
yao@calicolabs.com (YLW);
carmela@calicolabs.com (CS)

[†]These authors contributed equally to this work

## Introduction

Eukaryotic translation initiation factor 2B (eIF2B) is the essential heteropentameric guanine nucleotide exchange factor (GEF) for the GTPase eukaryotic translation initiation factor 2 (eIF2). eIF2 forms a ternary complex with GTP and the initiator methionyl tRNA (Met-tRNA$_i$). During each round of mRNA translation initiation, GTP is hydrolyzed, Met-tRNA$_i$ is incorporated into the nascent polypeptide and eIF2-GDP is released from the translating ribosome (reviewed in *Hinnebusch and Lorsch, 2012*). eIF2B is required to catalyze GDP-GTP exchange on eIF2, which enables rebinding of Met-tRNAi and a new round of protein synthesis.

In the context of normal physiology, transient inhibition of eIF2B GEF activity is a critical step in the activation of the conserved integrated stress response (ISR). In mammalian cells, four different stress-responsive kinases (PERK, GCN2, HRI and PKR) initiate the ISR by phosphorylating serine 51 in the α subunit of eIF2 (eIF2α). eIF2 in this phosphorylated state (eIF2α-P) binds more tightly to eIF2B and inhibits its activity resulting in rapid depletion of ternary complex and a reduction in bulk protein synthesis (*Krishnamoorthy et al., 2001*; *Yang and Hinnebusch, 1996*). The decrease in ternary complex paradoxically promotes translation of a subset of mRNAs, among them activating transcription factor 4 (ATF4). These transcripts escape the global inhibition of translation due to the presence of short upstream open reading frames (uORFs) in their 5' untranslated region (*Palam et al., 2011*; *Vattem and Wek, 2004*) Sheltered from the ISR, the expression of these genes during mild stress promotes cell survival. However, severe stress and sustained activation of this program can also promote pro-apoptotic pathways.

Given the central role of eIF2B and the ISR in modulating cell physiology, it is unsurprising that dysregulation of this pathway has pathological effects. One prominent example is Vanishing White

Matter Disease (VWMD), a rare, autosomal recessive leukodystrophy directly caused by mutations in eIF2B. VWMD is defined by chronic demyelination, with patients presenting with a spectrum of neurological symptoms including ataxia, spasticity, seizures and cognitive impairment (*Schiffmann et al., 1994*; *van der Knaap et al., 1997*). Age of disease onset is variable, and can be broadly categorized as early infantile, early childhood, late childhood/juvenile, and adult forms. The age of disease onset is correlated with severity and speed of progression, with younger cases being more severe (*Fogli et al., 2004a*; *Labauge et al., 2009*).

Since the discovery of the first VWMD causal mutation in eIF2B (*Leegwater et al., 2001*), more than 170 mutations spanning all five subunits of the eIF2B complex (α, β, γ, δ, ε) have been cataloged in the NCBI ClinVar database. Patients harbor either homozygous mutations, or display a compound heterozygous genotype, with a different mutation in each allele of the same eIF2B subunit gene. Because eIF2B activity is essential and all subunits are required in mammalian cells, mutations in the GEF domain of the ε catalytic subunit (which would be more likely to abrogate activity) are rare. Instead, eIF2B VWMD mutations reportedly cause a partial reduction in GEF activity (*Fogli et al., 2004b*; *Horzinski et al., 2009*; *Li et al., 2004*; *Liu et al., 2011*; *de Almeida et al., 2013*). Interestingly, while some experiments with patient-derived samples suggest a correlation between eIF2B activity and age of VWMD onset (*Fogli et al., 2004b*; *Horzinski et al., 2009*), others found no association with disease severity (*Liu et al., 2011*; *van Kollenburg et al., 2006a*). The interpretation of these experiments is complicated by the diverse genetic backgrounds of the samples. Thus, further work is needed to clarify the relationship between eIF2B GEF activity and VWMD.

Markers of ISR activation have been identified in VWMD brain tissue by immunostaining assays (*van Kollenburg et al., 2006b*). Since eIF2B GEF activity is constitutively reduced in patient cells, further inhibition by eIF2α-P could lead to persistent ISR induction and chronically reduced protein synthesis. This is consistent with the clinical description of VWMD pathology accelerating in response to stresses such as head trauma, fever and viral infection. Questions remain about the relative contributions of chronically reduced protein translation and stress pathway activation to white matter pathology, but recent work suggests that eIF2B VWMD mutations lead to defects in white matter astrocyte maturation (*Bugiani et al., 2011*; *Dooves et al., 2016*) and disrupt mitochondrial oxidative phosphorylation (*Raini et al., 2017*).

ISRIB (for ISR inhibitor) is a potent small molecule that attenuates the ISR by binding and activating eIF2B (*Sidrauski et al., 2013*, *2015*). The five subunits of eIF2B assemble into a decameric (αβδγε)$_2$ complex (*Kashiwagi et al., 2016*; *Wortham et al., 2014*). The homologous α, β and δ subunits form a hexameric α$_2$(βδ)$_2$ regulatory subcomplex containing the interaction site for eIF2α-P (*Bogorad et al., 2014*; *Kuhle et al., 2015*). The γ and ε subunits share a common ancestor and form a catalytic heterodimeric (γε) subcomplex. Two γε subcomplexes flank the core regulatory subcomplex to assemble the decameric holoenzyme. We proposed that ISRIB, a symmetrical molecule, stimulates eIF2B function by binding at the mirrored (βδ)$_2$ interface and promoting the assembly of the more active decameric complex (*Sidrauski et al., 2015*). In agreement with this model, mutations in the δ subunit were identified that make the complex insensitive to the action of ISRIB (*Kashiwagi et al., 2016*; *Sekine et al., 2015*). By enhancing eIF2B activity, ISRIB increases the levels of ternary complex, restoring protein synthesis and attenuating the induction of the ISR in cells.

Because eIF2B mutations cause VWMD, this disease is an ideal model to investigate pathogenic ISR activation and the effects of ISRIB in such a setting. However, it is unclear whether ISRIB can interact with the wide range of eIF2B mutant complexes found in VWMD patients, and if so, whether it can rescue their residual GEF activity to sufficient levels. To measure the effect of VWMD mutations on GEF activity and their responsiveness to ISRIB, we deployed two complementary approaches: biochemical assays with reconstituted mutant eIF2B complexes, and cell-based assays utilizing CRISPR-Cas9 to introduce various VWMD mutations in an isogenic background. We found that VWMD mutations destabilize the decameric eIF2B holoenzyme, directly correlating with GEF activity impairment. Addition of ISRIB stabilized recombinant mutant eIF2B complexes in the decameric form, enhancing their residual GEF activity to wild-type levels. ISRIB also restored the GEF activity of lysates derived from mutant cells to the level of the isogenic wild-type. In all mutant cells, ISRIB significantly attenuated the two outputs of the ISR: the global reduction in protein synthesis,

and the induction of ATF4. Collectively, our data demonstrate that VWMD mutations in eIF2B that compromise its enzymatic activity do so by disfavoring decameric complex formation, and ISRIB rescues a wide range of mutations by stabilizing a more active decameric complex irrespective of the nature of the crippling mutation.

## Results

### A fully reconstituted system to assay the GEF activity of eIF2B complexes

In order to isolate the effect of pathogenic VWMD mutations on eIF2B GEF activity, we first developed an assay using purified recombinant human complexes. A minimal defined system requires substrate (heterotrimeric eIF2 complex), enzyme (heteropentameric eIF2B complex) and a method of monitoring reaction kinetics. We expressed and purified recombinant human eIF2 from human cells (*Figure 1A and B*), and loaded it with a fluorescent GDP analog, Bodipy-FL-GDP. The fluorescence quantum yield of this analog is increased upon protein binding and quenched in solution; release of Bodipy-FL-GDP from eIF2 is thus measured as a time-dependent decrease in fluorescence (*Sekine et al., 2015*). In the presence of excess unlabeled GDP, the fluorescence signal of Bodipy-FL-GDP follows a pattern of single-exponential decay. As previously reported, eIF2 alone exhibits a slow intrinsic rate of GDP exchange ($t_{1/2}$ > 70 mins; *Figure 1C*; *Panniers and Henshaw, 1983*).

Based on crystallographic and mass spectrometric data (*Kashiwagi et al., 2016*; *Kuhle et al., 2015*; *Wortham et al., 2014*), the βδγε subunits of eIF2B form a stable heterotetrameric subcomplex in vitro, which can be further 'dimerized' by association with an $\alpha_2$ subcomplex to form the decameric holoenzyme. We co-expressed and affinity-purified either all five wild-type eIF2B subunits together from human cells, or the βδγε and α subunits separately (*Figure 1D*). We reasoned that the first approach would allow us to selectively obtain pre-formed eIF2B decameric complexes, whereas the second approach would allow us to reconstitute the process of holoenzyme assembly in vitro.

Co-expression of five eIF2B subunits yielded a purified complex with a size corresponding to the predicted $(\alpha\beta\delta\gamma\epsilon)_2$ decamer, as determined by multi-angle light scattering (*Figure 1E*). Expression of the α subunit by itself produced a homodimer, whereas a heterotetramer was formed by βδγε in the absence of α (*Figure 1E*). Thus, the purification process yielded intact subcomplexes containing equimolar quantities of each eIF2B subunit. Addition of purified eIF2B decamer to eIF2 stimulated GDP release and increased the rate of fluorescence decay ($t_{1/2}$ = 20 mins; *Figure 1C*). Consistent with prior results, the stimulatory effect of eIF2B on eIF2 GDP release was further enhanced by addition of ISRIB in a dose-dependent manner (*Sekine et al., 2015*; *Sidrauski et al., 2015*). The combination of βγδε and $\alpha_2$ behaved similarly to the pre-formed decameric complex (see below).

The eIF2B δ subunit exhibits alternative splicing, producing two isoforms, δ1 and δ2, which differ only at their N-terminal region (*Figure 1—figure supplement 1A*; *Kuhn et al., 2009*). Prior work in cells suggested that the longer and less-studied δ1 isoform may render eIF2B insensitive to inhibition by eIF2α phosphorylation by weakening the eIF2-eIF2B interaction (*Martin et al., 2010*). As a first step, we used our reconstituted system as a platform to compare the two different δ isoforms. We purified decamers of both isoforms (*Figure 1D*) to compare their effectiveness at stimulating GDP release. Additionally, we stoichiometrically phosphorylated trimeric eIF2 at serine 51 of its α subunit using recombinant PERK kinase, to serve as an eIF2B inhibitor in the GEF assay.

We found that the δ1 and δ2 isoforms had similar basal GEF activity and responded similarly to ISRIB stimulation ($EC_{50}$ = 0.8 nM and 1.2 nM, respectively; *Figure 1F*). Next, while keeping the total eIF2 concentration constant, we titrated phospho-eIF2 into the system to a maximum of two-fold excess over unphosphorylated eIF2. As expected, the GDP release $t_{1/2}$ increased asymptotically with addition of phospho-eIF2 (*Figure 1G*), approaching the intrinsic rate of eIF2 GDP release without eIF2B. In contrast to previous reports, we observed no difference between the two δ isoforms in their response to phospho-eIF2. Notably, we showed that ISRIB equally reduced the inhibitory effect of phospho-eIF2 on both isoforms, such that even a 2:1 ratio of inhibitor:substrate was insufficient to block eIF2B activity. We conclude that there is no difference between the two δ isoforms under our experimental conditions.

The results above validated our assay as a sensitive and reproducible method to monitor the GEF activity of recombinant eIF2B complexes. Based on these measurements, we generated VWMD

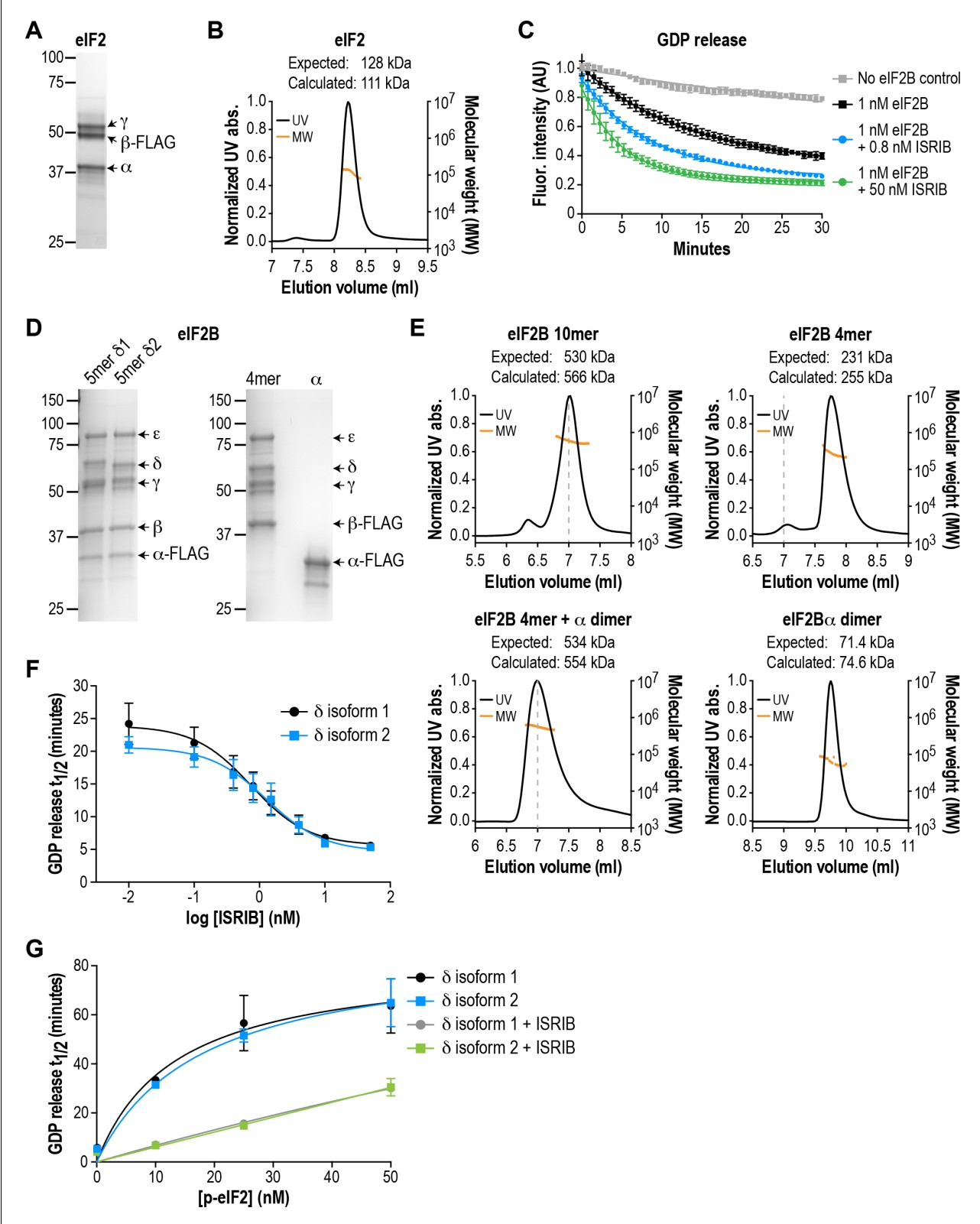

**Figure 1.** eIF2B isoforms have identical GEF activity and ISRIB alters the sensitivity of eIF2B to eIF2 phosphorylation. (**A**) Coomassie-stained SDS-PAGE gel of recombinant human eIF2 used as a substrate in the in vitro GEF assays. (**B**) Multi-angle light scattering analysis of purified eIF2, indicating good agreement between measured and expected molecular weights for the heterotrimeric complex. (**C**) eIF2B stimulates GDP release from eIF2 that is further enhanced by addition of ISRIB. Bodipy-FL-GDP was loaded onto eIF2, and its release measured over time as a decay in fluorescence with or

*Figure 1 continued on next page*

*Figure 1 continued*

without eIF2B and ISRIB. Representative fluorescence intensity curves are shown. Intensities were normalized to the starting values for the no-eIF2B control condition. (D) Coomassie-stained SDS-PAGE gels of purified recombinant human eIF2B complexes. (E) Multi-angle light scattering analysis of purified WT eIF2B. The $(\alpha\beta\gamma\epsilon)_2$ complex ($\delta2$ isoform) ran as a 10mer. The $\beta\delta\gamma\epsilon$ subcomplex ran as a 4mer, whereas the $\alpha$ subunit forms a homodimer. Combining separately purified $\beta\delta\gamma\epsilon$ and $\alpha$ subunit reconstituted the 10mer. The dashed gray line marks the 7 ml elution position to facilitate comparisons between plots. (F) $\delta1$ and $\delta2$ isoforms of eIF2B behaved identically with respect to GEF activity and response to ISRIB. 5 nM phospho-eIF2 was included in the assay. (G) Dose-response curves of GDP release half-life in the presence of increasing amounts of phospho-eIF2. The curves of $\delta1$ + ISRIB (*grey*) and $\delta2$ + ISRIB (*green*) overlap each other. For (F) and (G), half-lives of GDP release at each ISRIB concentration were calculated by fitting single-exponential decays to the Bodipy-FL-GDP fluorescence decay curves. Each point represents 9 measurements from three independent experiments (mean ± SD).

DOI: https://doi.org/10.7554/eLife.32733.002

The following figure supplement is available for figure 1:

**Figure supplement 1.** Purification of eIF2B mutants.

DOI: https://doi.org/10.7554/eLife.32733.003

mutants in the background of eIF2B with the $\delta2$ isoform, which is the predominant isoform in vivo (*Martin et al., 2010*). All subsequent experiments also included a small fraction of unlabeled phospho-eIF2 (5 nM added to 25 nM eIF2:Bodipy-FL-GDP) in the assays to increase the dynamic range of the response to ISRIB stimulation.

## Recombinant eIF2B with VWMD mutations exhibit impaired GEF activity that is rescued by ISRIB

With an in vitro measurement system in place, we next examined the effects of pathogenic VWMD mutations on the GEF activity of eIF2B. Using the structure of *S. pombe* eIF2B as a guide, we selected five VWMD mutations that vary in their location within the eIF2B complex and the severity of the resulting disease (*Figure 2A* and *Table 1*). Two of the mutations reside in the $\alpha\beta\delta$ regulatory subcomplex. $\alpha$V183F, located at the interface of the $\alpha$-$\alpha$ homodimer, has been shown to abolish dimer formation (*Wortham et al., 2014*). Similarly, $\delta$R483W is predicted to disrupt a key $\beta$-$\delta$ heterodimer interaction that is important for tetramerization of the $(\beta\delta)_2$ subcomplex. R113 in the catalytic $\epsilon$ subunit contacts the $\beta$ subunit and the mutation of Arg to His likely destabilizes the interaction between the catalytic and regulatory subcomplexes (*Kashiwagi et al., 2016*). $\epsilon$R113H is one of the most prevalent VWMD mutations (*Table 1*). We also generated $\epsilon$R136H and $\epsilon$R195H mutations, which are surface-exposed and buried residues, respectively. Patients with the $\epsilon$R136H mutation exhibit classical disease progression (*Kantor et al., 2005*), whereas the $\epsilon$R195H mutation results in a severe, early-onset form of VWMD known as Cree leukoencephalopathy (*Black et al., 1988*; *Fogli et al., 2002*). Notably, animal models have been generated for the $\delta$R483W, $\epsilon$R136H and $\epsilon$R195H mutations, and they recapitulate important aspects of the human disease (*Dooves et al., 2016*; *Geva et al., 2010*).

To better control for the effects of these mutations on eIF2B holoenzyme formation, we expressed them as separate $\beta\gamma\delta\epsilon$ and $\alpha_2$ subcomplexes and combined equimolar amounts of each subcomplex for the GEF assay. Similar to wild-type proteins, all the mutant subcomplexes purified to homogeneity with similar yield (*Figure 1—figure supplement 1B*). Wild-type eIF2B promoted GDP release from eIF2 with $t_{1/2}$ = 12.2 mins (*Figure 2B* and *Figure 2—figure supplement 1A*). Under identical conditions, every VWMD mutant except $\epsilon$R136H had significantly slowed GDP release. $\alpha$V183F had 33% of wild-type basal activity, whereas $\delta$R483W, $\epsilon$R113H and $\epsilon$R195H had 28%, 61% and 56% activity, respectively. Thus, 4 out of 5 VWMD mutations tested in this fully reconstituted system exhibited compromised GEF activity. The wild-type GEF activity of the $\epsilon$R136H mutant complex is intriguing. It is possible that the mutation destabilizes the endogenous $\epsilon$ subunit in cells. However, no reduction in the affected subunit was observed in cerebellar lysates from mice carrying this mutation and a 23% reduction in GEF activity was reported (*Geva et al., 2010*).

In our assay, ISRIB stimulated the activity of wild-type eIF2B ~ 2X in a dose-dependent manner ($EC_{50}$ = 0.9 nM; *Figure 2B and C*; *Figure 2—figure supplement 1A*). 50 nM ISRIB successfully rescued the activity of the VWMD mutants, with stimulation ranging from 1.8X for $\alpha$V183F up to 7.4X for $\delta$R483W (*Figure 2B*). Indeed, the ISRIB-rescued GEF activity of every mutant except $\alpha$V183F exceeded the level of basal wild-type activity. The similar ISRIB $EC_{50}$ values across wild-type and

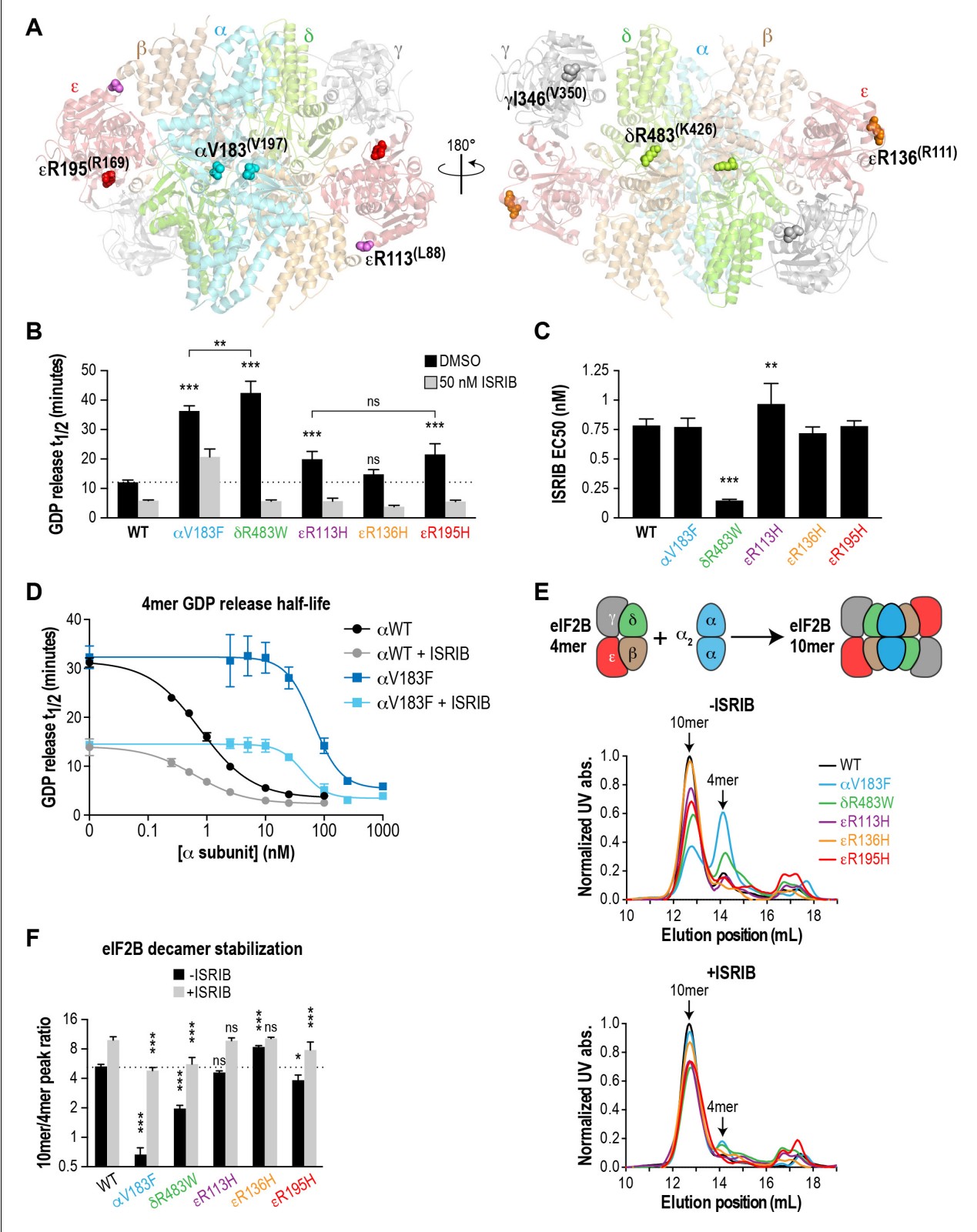

**Figure 2.** Recombinant eIF2B VWMD mutants have reduced GEF activity and complex stability that is rescued by ISRIB. (**A**) VWMD mutations tested in this study, visualized as spheres on the structure of *S. pombe* eIF2B (PDB: 5B04; *Kashiwagi et al., 2016*). The human mutation sites are shown in plain text and the corresponding yeast residues in superscript. For clarity, only one of each pair of residues is labeled. (**B**) Half-lives of GDP release for recombinant WT and VWMD eIF2B, calculated by fitting single-exponential decays to the Bodipy-FL-GDP fluorescence decay curves. All VWMD
*Figure 2 continued on next page*

*Figure 2 continued*

mutants except εR136H had reduced GEF activity that is significantly stimulated by ISRIB (p<0.001). Significance is shown for comparisons of each mutant to WT and between selected mutants. **p<0.005, ***p<0.001, ns, not significant. (C) ISRIB EC$_{50}$ values calculated from dose-response measurements in GEF assays (individual curves shown in *Figure 2—figure supplement 1A*). Significance is shown for comparisons of each mutant to WT. **p<0.01, ***p<0.001. (D) Dose-response curves of GDP release half-life in the presence of increasing amounts of WT or V183F α subunit. For (B)-(D), half-lives of GDP release at each ISRIB/α concentration were calculated by fitting single-exponential decays to the Bodipy-FL-GDP fluorescence decay curves. Each point represents 9 measurements from three independent experiments (mean ±SD). (E) All eIF2B VWMD mutant complexes except εR136H have reduced complex stability compared to WT. Size-exclusion chromatograms of reconstituted wild-type and VWMD mutant eIF2B complexes in the presence of 300 mM NaCl. The elution positions of the (αβδγε)$_2$ 10mer and βδγε 4mer are indicated. A cartoon schematic of 10mer formation is shown above the graphs. (F) Quantification of eIF2B 10mer:4mer peak ratios from (E), which serves as a measure of complex stability (N = 3, mean ± SD). *p<0.05, ***p<0.001, ns, not significant. Significance is shown for comparisons to WT within a condition (vehicle or ISRIB). Between conditions for a given construct, all differences were significant (p<0.005).

DOI: https://doi.org/10.7554/eLife.32733.004

The following figure supplements are available for figure 2:

**Figure supplement 1.** Recombinant eIF2B VWMD mutants have reduced GEF activity that is rescued by ISRIB.

DOI: https://doi.org/10.7554/eLife.32733.005

**Figure supplement 2.** Recombinant eIF2B VWMD mutants have reduced complex stability that is rescued by ISRIB.

DOI: https://doi.org/10.7554/eLife.32733.006

---

VWMD mutant complexes (*Figure 2C*) indicate that the mutations do not affect the ability of ISRIB to bind to eIF2B and stimulate its GEF activity. In the case of the δR483W mutant, the ISRIB EC$_{50}$ was lower than wild-type, indicating that ISRIB was actually more effective at enhancing the activity of this mutant.

αV183F is unique among the mutations tested in this study because it is the only one that affects the α$_2$ homodimer instead of the βδγε subcomplex. Unlike the other mutations, ISRIB was unable to normalize its activity to wild-type levels even though the EC$_{50}$ was unchanged (*Figure 2B and C*). To further characterize this mutation, we performed a GEF assay with wild-type βδγε and titrated α$^{WT}$ or α$^{V183F}$. It has been shown that the heterotetramer containing the catalytic eIF2Bε subunit retains a low level of GEF activity that is enhanced by supplementation of eIF2Bα (*Williams et al., 2001*). Consistent with this, we observed that the activity of βδγε was <30% of decamer (*Figure 2D*, compare left-most point of black curve against points with >2.5 nM eIF2Bα). βδγε activity was stimulated 2.3X in the presence of 50 nM ISRIB (*Figure 2D*, compare left-most points of black and gray curves),

---

**Table 1.** eIF2B mutants generated in this study

| eIF2B subunit | Mutation | Age of disease onset (years) | Disease alleles | HEK293T cells generated | Recombinant protein generated |
|---|---|---|---|---|---|
| **α** | V183F | 10–17 | Homozygous* | X | X |
| **γ** | I346T | 1–4 | Homozygous; also compound heterozygous with G47E[†] | X | - |
| **δ** | R483W | <1 | Homozygous[‡] | X | X |
| **ε** | R113H | 1–30 | Homozygous; also compound heterozygous with multiple other mutations[§] | X | X |
| **ε** | R136H | 3 | Homozygous[#] | - | X |
| **ε** | R195H | <1 | Homozygous** | X | X |
| **δ** | L179F | ISRIB-resistant; not naturally occurring[††] | | - | X |
| **δ** | L487W | ISRIB-resistant; not naturally occurring | | - | X |

*Ohlenbusch et al., 2005

[†]Wu et al., 2009

[‡]van der Knaap et al., 2003

[§]Fogli et al., 2004

[#]Kantor et al., 2005

**Fogli et al., 2002

[††]Sekine et al., 2015

DOI: https://doi.org/10.7554/eLife.32733.007

concomitant with stabilization of a more-active octameric form (*Figure 2D* and *Figure 2—figure supplement 1B*). Addition of α subunit increased GEF activity in a concentration-dependent manner, with $K_m$ = 6 nM for $α^{WT}$ and 246 nM for $α^{V183F}$; 40X more mutant subunit was required to achieve the same level of GEF activity compared to wild-type (*Figure 2—figure supplement 1C*). Thus, dimerization of the α subunit, which is abolished by the V183F mutation, provides a critical avidity effect that promotes eIF2B complex formation and activity. ISRIB shifted the $K_m$ of $α^{WT}$ and $α^{V183F}$ to 4 nM and 88 nM, respectively. The assays in *Figure 2C* were performed with 3 nM eIF2B, which explains the incomplete rescue of αV183F mutant activity by ISRIB; under those conditions, α concentration was still 30X below the αV183F $K_m$. We show that using sufficiently high concentrations of $α^{V183F}$, wild-type levels of GEF activity are achievable, both in the absence and presence of ISRIB (*Figure 2D*).

In conclusion, we show that four different VWMD mutant eIF2B complexes display deficient GEF activity in a fully reconstituted system. ISRIB enhanced the activity of all generated mutants, even in a case where the biochemical consequence of the disease-causing mutation (εR136H) could not be readily assessed.

## Recombinant eIF2B with VWMD mutations exhibit reduced decamer stability that is rescued by ISRIB

Having established that VWMD mutant complexes show compromised GEF activity, we wondered how this impairment was elicited by the various mutations that affect different subunits. A clue came from comparing the activity of tetrameric (-$α_2$ dimer) and decameric (+$α_2$ dimer) eIF2B; the former is significantly poorer at stimulating GDP release (*Figure 2D*; compare activity of tetramer before and after addition of α subunit). We postulated that some VWMD mutations might destabilize the eIF2B holoenzyme, thus leading to reduced activity. To examine the ability of eIF2B complexes to form decamers, we combined the purified βδγε and $α_2$ subcomplexes, and subjected the mixtures to size-exclusion chromatography (SEC). We used an elevated salt concentration (300 mM) to destabilize the eIF2B complex slightly and increase the dynamic range of the assay, allowing us to interrogate the effect of the pathogenic mutations as well as the ability of ISRIB to enhance decamer formation. A similar approach was previously used to demonstrate the stabilizing effect of ISRIB on eIF2B-containing lysates by sucrose gradient (*Sidrauski et al., 2015*).

Wild-type eIF2B readily formed a decamer as the major species, eluting at 12.7 mL on the SEC column (*Figure 2E* and *Figure 2—figure supplement 2A*). A minor peak corresponding to the βδγε tetramer was observed at 14.1 mL, whereas the $α_2$ homodimer eluted at 16.7 mL. By calculating the ratio of peak areas for decamer:tetramer ($R_{10/4}$), we obtained a simple metric for complex stability (*Figure 2F*). The wild-type complex had a $R_{10/4}$ of 5.2 under these assay conditions. Remarkably, αV183F, δR483W, εR113H and εR195H had decreased $R_{10/4}$s of 0.6, 2.0, 4.6 and 4.0, respectively, demonstrating that these pathogenic mutations destabilize the decameric eIF2B complex (*Figure 2E–F* and *Figure 2—figure supplement 2A*). The εR136H mutant had a $R_{10/4}$ of 8.5, indicating decamer stability as good as, if not better than wild-type.

We repeated the SEC experiment, but with the mobile phase supplemented with 200 nM ISRIB. In agreement with our hypothesis, we observed a collapse of the tetramer peak and an increase in the decameric species for all eIF2B constructs (*Figure 2E* and *Figure 2—figure supplement 2A*). This change was particularly large for the αV183F mutant, which underwent an 8.2X increase in $R_{10/4}$ to 4.9 after ISRIB addition, which is comparable to that of wild-type complex in the absence of drug (*Figure 2F*).

Our data show that decamer formation is presumably correlated with eIF2B function (GEF activity of wild-type ~ εR136H > εR113H ~ εR195H > αV183F > δR483W). The αV183F mutation produced a particularly severe effect on complex stability, but the GEF activity of the mutant is better than expected because its βδγε subcomplex is wild-type. By contrast, the other mutants fall into a distinct class wherein defects affect the heterotetrameric subcomplex directly. Our results suggest that VWMD mutations that impair GEF activity may do so via a common mechanism of destabilizing the decameric eIF2B holoenzyme. This could explain why compound heterozygote alleles lead to VWMD; there are many non-targeted ways to push a complex away from optimal stability. The existence of over 170 eIF2B mutations in human VWMD likely reflects the ease with which the multi-subunit eIF2B assembly can be disrupted. To our knowledge, this is the first demonstration that ISRIB

can engage with mutant eIF2B complexes, stabilize formation of the highly active decamer, and rescue GEF activity in vitro.

## ISRIB-resistant mutants with wild-type or VWMD-like properties validate the correlation between decamer stability and GEF activity

As a further test of our hypothesis that eIF2B decamer formation drives enhanced GEF activity, we wondered whether we could generate a synthetic VWMD-like mutation that would also be resistant to ISRIB stimulation by mutating residues in the vicinity of the ISRIB binding pocket. eIF2Bδ is a key mediator of ISRIB binding, based on unbiased activity-based screening (*Sekine et al., 2015*) and biochemical stabilization assays (*Sidrauski et al., 2015*). Modeling of the ISRIB-resistant mutations from the activity-based screen onto the *S. pombe* eIF2B structure placed them near the pseudo-twofold rotational axis of eIF2B (*Kashiwagi et al., 2016*), consistent with the proposed mechanism of stabilization. Examination of the eIF2B crystal structure revealed a cavity at its core that could potentially accommodate ISRIB; the ISRIB-resistant mutants form part of the surface of this space (*Kashiwagi et al., 2016*). To date, no VWMD mutations have been identified in this immediate region. We identified another cavity-facing residue in the δ subunit, L487, which also packs against the adjacent β subunit (*Figure 3A*). We reasoned that introducing a bulky residue at the L487 position might simultaneously disrupt both eIF2B complex stability and ISRIB binding.

We generated the δL487W mutation in the background of the eIF2B $(\alpha\beta\delta\gamma\epsilon)_2$ decamer, as we were unable to purify it in a tetrameric form. As a control, we also generated δL179F, which was identified in an unbiased screen as an ISRIB-resistant mutant with wild-type activity (*Sekine et al., 2015*). First, we compared decamer stability of the wild-type and mutants by SEC on the purified $\alpha\beta\delta\gamma\epsilon$ complexes. ISRIB supplementation increased the $R_{10/4}$ of wild-type eIF2B from 8.6 to 15.5 (*Figure 3B–C* and *Figure 2—figure supplement 2B*). The δL487W mutation had a markedly reduced $R_{10/4}$ of 1.6 (6X lower than wild-type) that did not change significantly in the presence of ISRIB, confirming that we had generated an ISRIB-resistant mutant. As expected, the δL179F also did not respond to ISRIB, although its $R_{10/4}$ of 5.6 was closer to wild-type eIF2B (*Figure 3B–C* and *Figure 2—figure supplement 2B*).

Like bona-fide VWMD mutations that destabilize the decameric complex, we expected δL487W to have impaired GEF activity. Consistent with the SEC data, the GEF activity of the δL487W mutant was severely compromised (25% of wild-type activity), whereas the activity of the δL179F mutant was similar to wild-type (*Figure 3D*) and both mutants were unresponsive to the addition of up to 500 nM ISRIB (550X wild-type $EC_{50}$). Finally, we took advantage of a purified δL179F $\beta\delta\gamma\epsilon$ ISRIB-resistant tetramer, and combined it with the αV183F VWMD mutation. As expected, the combination of both mutations displayed not only compromised GEF activity which is driven by the αV183F mutation, but an inability to be rescued by ISRIB (*Figure 3E*). Thus, our examination of existing structural knowledge allowed us to generate a novel ISRIB-resistant mutant with VWMD-like complex stability and activity, which we contrasted against another ISRIB-resistant mutant with wild-type properties.

## Several VWMD mutations reduce the protein level of the affected subunit in cells

The positive effects of ISRIB in vitro motivated us to generate a cellular model of VWMD, which would allow us to test its ability to rescue functional outputs of the ISR. We aimed to introduce mutations into an isogenic mammalian cell background in order to make direct comparisons between cell lines. We first isolated a monoclonal cell line from a previously described polyclonal HEK293T population containing an ISR reporter stably integrated at multiple genomic sites (*Sidrauski et al., 2013*). The ISR reporter drives firefly luciferase expression under control of the ATF4 5′ untranslated region (containing two uORFs). Like endogenous ATF4, translation of this reporter is driven by stress-induced eIF2α phosphorylation, which reduces ternary complex formation and results in translation initiation in the luciferase open reading frame (*Sidrauski et al., 2013*). We reasoned that this reporter would enable us to quantitatively measure ISR induction in cells carrying the various eIF2B VWMD mutations.

Using the monoclonal parental ISR reporter cells, we generated daughter cell lines individually harboring the αV183F, δR483W, εR113H and εR195H mutations at their endogenous loci by CRISPR-Cas9 editing and homology-directed repair (*Figure 4—figure supplement 1A*). Despite repeated

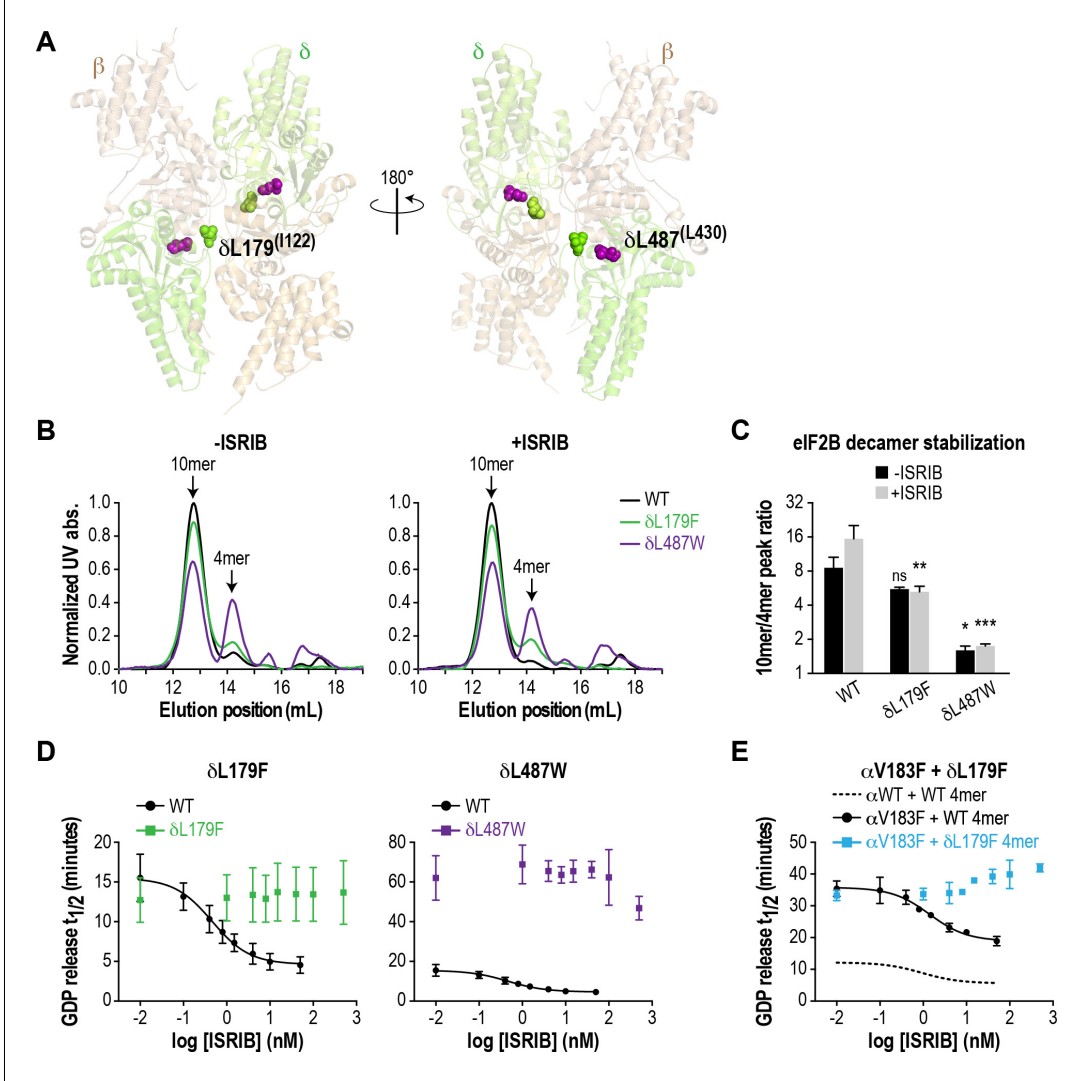

**Figure 3.** ISRIB-resistant mutants in the δ subunit can exhibit WT or VWMD-like stability and GEF activity. (A) ISRIB-resistant mutations tested in this study, visualized as spheres on the structure of *S. pombe* eIF2B (PDB: 5B04; *Kashiwagi et al., 2016*). The human mutation sites are shown in plain text and the corresponding yeast residues in superscript. For clarity, only one of each pair of residues is labeled. (B) ISRIB-resistant mutants have reduced complex stability and are unresponsive to 200 nM ISRIB. Size-exclusion chromatograms of reconstituted WT and ISRIB-resistant eIF2B complexes in the presence of 300 mM NaCl. The elution positions of the $(\alpha\beta\delta\gamma\epsilon)_2$ 10mer and $\beta\delta\gamma\epsilon$ 4mer are indicated. (C) Quantification of eIF2B 10mer:4mer peak ratios from (A) (N = 2, mean ±SD). *p<0.05, **p<0.005, ***p<0.001, ns, not significant. Significance is shown for comparisons to WT within a condition (vehicle or ISRIB). Between conditions for a given construct, only WT was significant with p<0.05. (D) The GEF activity of ISRIB-resistant mutants is not enhanced by up to 500 nM ISRIB (550X WT EC$_{50}$). ISRIB dose-response curves of GDP release half-life for WT, δL179F and δL487W 5mers. (E) ISRIB dose-response curves of GDP release half-life for WT and δL179F 4mer mixed with αV183F. Data from the WT 4mer + αWT is reproduced from *Figure 2—figure supplement 1A* as a dotted line for comparison. For (D) and (E), each point represents 9 measurements from three independent experiments (mean ± SD).

DOI: https://doi.org/10.7554/eLife.32733.008

attempts, we were unable to produce cells with the εR136H mutation. As a substitute, we generated a 'classical VWMD' I346T mutation in the γ subunit, the second subunit of the catalytic subcomplex. For each VWMD mutation, we isolated and validated two independent clones. Due to the low efficiency of homology-directed repair and the hypotriploid nature of HEK293T cells (three copies of each eIF2B gene), we only obtained true homozygous VWMD mutations for αV183F (2 clones), δR483W (1 clone) and εR195H (1 clone). The remaining clones carried one or two VWMD alleles, with the remaining allele(s) disrupted by an out-of-frame indel leading to a premature stop codon

(*Supplementary file 1*). Nevertheless, the VWMD allele is the only source of the affected subunit for all clones analyzed in this study. Thus, these cell lines do not suffer from the complication of residual wild-type eIF2B activity.

Previous studies in both yeast and mammalian cells noted that a subset of VWMD mutations reduced levels of the affected subunit, and in some instances also destabilized associated subunits of the mutant complex (*Richardson et al., 2004*; *Sekine et al., 2016*; *Wortham et al., 2016*). We anticipated that decreasing subunit levels might further cripple the activity of VWMD mutant eIF2B, above and beyond the effects on decamer stability established above. To determine whether the generated VWMD mutations decrease the level of the various subunits, we performed immunoblotting of each eIF2B polypeptide in wild-type and VWMD mutant cells. The αV183F mutation caused a consistent 50% reduction in the level of the α subunit in both clones generated, whereas the γ subunit showed a 50% reduction in only one clone (*Figure 4A*; *Figure 4—figure supplement 1B–C*). The γI346T mutation caused a 20% reduction (that did not reach statistical significance) in γ subunit in one clone, and a significant 90% reduction in the second clone. Both the εR113H and εR195H mutations produced significant 20–40% reductions in ε subunit levels. One clone of εR195H also showed a 40% reduction in the α, β and δ regulatory subcomplex (*Figure 4—figure supplement 1B*). As noted in previous studies, VWMD mutations negatively affected the abundance of the mutated subunit, and in a subset of clones, the level of additional subunits was also reduced. Because both αV183F clones and one of the εR195H clones are homozygous mutants, the effect on protein levels cannot be explained solely by reduced gene dosage. The observed differences between clones carrying the same mutation could possibly be due to the intrinsic genomic instability of HEK293T cancer cells. In the subsequent experiments, we analyzed both clones of each VWMD mutation.

## ISRIB attenuates ISR induction in all VWMD cell lines

Loss of eIF2B function is expected to produce a chronic ISR signal, and both VWMD patient-derived fibroblasts and post-mortem brain samples have markers of ISR induction (*Kantor et al., 2005*) (*van der Voorn et al., 2005*). Thus, we assumed that our cell lines with introduced VWMD mutations would exhibit a basal signature of ISR induction. To test this, we measured the two proximal outputs of the ISR: increased levels of ATF4 protein and attenuation of global protein synthesis (*Figure 4B*).

First, we compared the level of endogenous ATF4 under steady-state, unstressed conditions. We did not detect significant basal ATF4 expression in any of the VWMD mutants (*Figure 4C* and *Figure 4—figure supplement 2A*, first lane in each panel), even in the mutants with reduced eIF2B subunit levels; two clones, δR483W clone #2 and εR113H clone #2, showed weak expression that was not reproducible. ATF4 induction is a highly sensitive readout of ISR activation, thus the lack of upregulation in the mutants was unexpected. Next, we measured $^{35}$S-methionine incorporation to determine the relative rates of protein synthesis in the various lines. Consistent with a lack of ATF4 upregulation, the basal rate of protein synthesis was similar in both VWMD and wild-type cell lines (*Figure 4—figure supplement 2B–C*). These data are consistent with prior reports showing that in two other systems, VWMD patient-derived lymphoblasts and engineered VWMD mutant CHO cells, the rate of protein synthesis was not reduced (*Sekine et al., 2016*; *van Kollenburg et al., 2006a*). We conclude that HEK293T VWMD mutants do not show significant ISR activation under basal conditions, suggesting that eIF2B activity is not limiting in this context. This is likely due to the high level of eIF2B expression in transformed cells (*Figure 4—figure supplement 4A–B*; *Balachandran and Barber, 2004*; *Kim et al., 2000*), which would mask the effects of VWMD mutations. Increased levels of eIF2B ensure that transformed cells can maintain high rates of protein synthesis for rapid proliferation.

We wondered whether challenging cells with an exogenous stressor might reveal differences in ISR induction between wild-type and VWMD cells. To this end, we treated cells with tunicamycin (Tm), an inhibitor of protein N-glycosylation that triggers the ISR through activation of the PERK kinase. The level of endogenous ATF4 in wild-type and VWMD cells was significantly increased after 3 hr of Tm treatment (*Figure 4C* and *Figure 4—figure supplement 2A*, second lane in each panel). Importantly, co-treatment with 500 nM ISRIB fully suppressed the upregulation of ATF4 protein in wild-type as well as VWMD cell lines (*Figure 4C* and *Figure 4—figure supplement 2A*, third lane in each panel). This result demonstrates that even though we did not observe basal ISR induction under

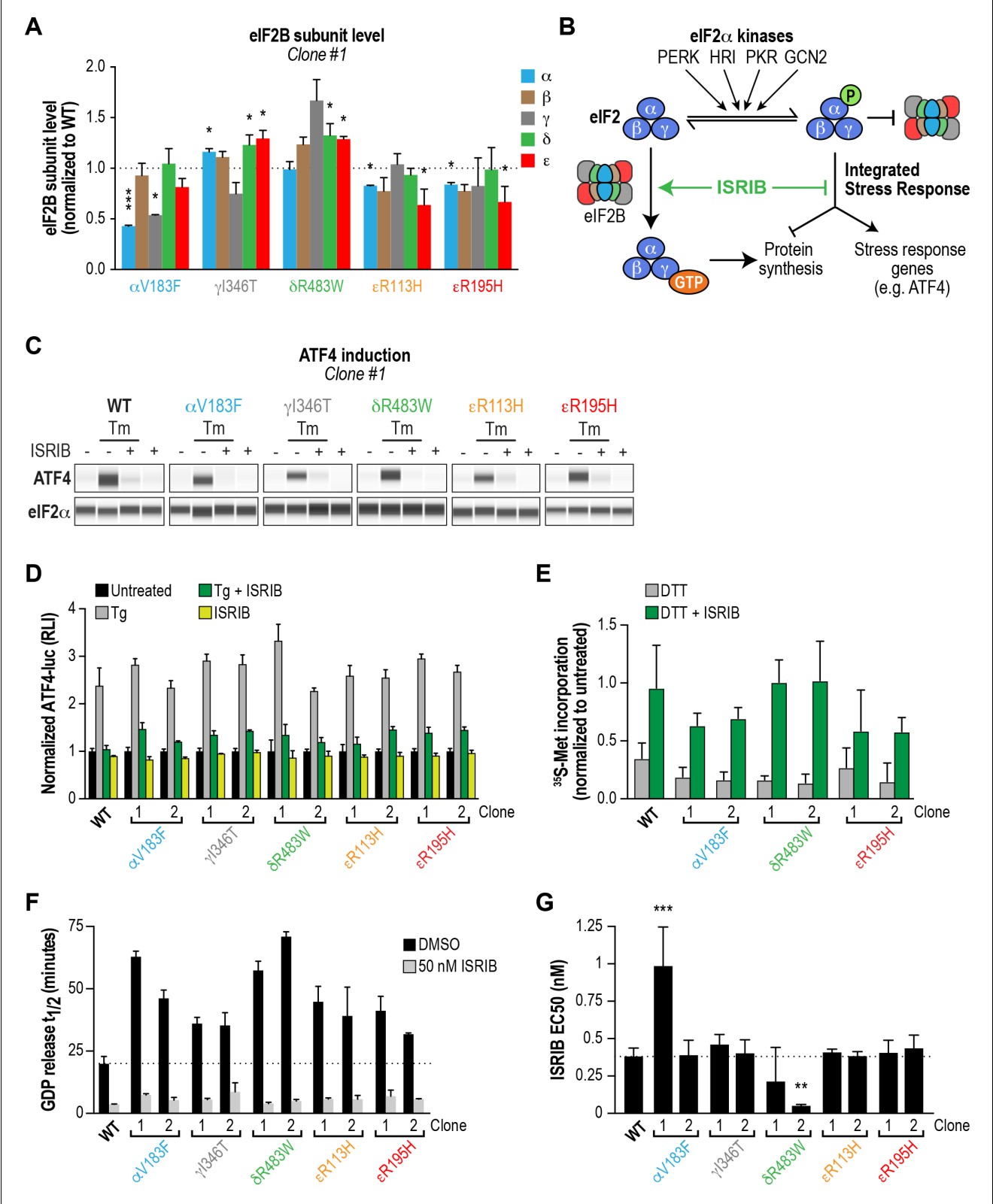

**Figure 4.** ISRIB attenuates the ISR and rescues the GEF activity of eIF2B in HEK293T VWMD mutant cells. (**A**) Levels of the five eIF2B subunits were measured by immunoblotting (Wes analysis) of lysates from WT and VWMD cell lines. The chemiluminescence signal of each subunit was normalized to α-tubulin as a loading control and subsequently normalized again to WT subunit levels (N = 2, mean ± SD). The Wes image of one replicate is shown in *Figure 4—figure supplement 1B*. Quantification of a second clone of each mutant is shown in *Figure 4—figure supplement 1C*. Significance is

*Figure 4 continued on next page*

*Figure 4 continued*

shown for each subunit compared to WT; *p<0.05, ***p<0.001. (**B**) Schematic depicting the two outputs of the ISR: reduction in bulk protein synthesis and translational induction of ISR target genes. (**C**) ISRIB blocks ATF4 induction in VWMD mutants. Immunoblot analysis of endogenous ATF4 protein levels in WT and VWMD mutant cells. Total eIF2α was used as a loading control. Cells were treated with or without 1 μg/mL Tm and 500 nM ISRIB for 3 hr. A representative experiment is shown (N = 2). Similar results were observed in the second clone of each mutant (**Figure 4—figure supplement 2A**). (**D**) ISRIB attenuates Tg-evoked reporter activity in VWMD mutants. Relative luminescence values for the indicated cell lines treated with 100 nM Tg ± 500 nM ISRIB for 3 hr (N = 4, mean ± SD). The values were normalized to the signal of untreated control within each cell line. In every cell line, ISRIB significantly reduced the Tg-induced reporter signal (p<0.001). (**E**) ISRIB rescues $^{35}$S-methionine incorporation in both WT and VWMD mutants to a similar extent. Cells were treated with 2.5 mM DTT ± 500 nM ISRIB for 1 hr and pulsed with $^{35}$S-methionine 20 min prior to collection. Lysates were subjected to SDS-PAGE and quantification of radiolabeled methionine incorporation was done by gel densitometry in a Typhoon FLA scanner and normalized to that of untreated cells (N = 2, mean ± SD). Raw images used to quantify the lane intensities are shown in **Figure 4—figure supplement 2D**. (**F**) Half-lives of GDP release for WT and VWMD HEK293T lysates, calculated by fitting single-exponential decays to the Bodipy-FL-GDP fluorescence decay curves. All VWMD mutants had significantly reduced GEF activity (p<0.001) that is stimulated by ISRIB. (**G**) ISRIB $EC_{50}$ values calculated from dose-response measurements in GEF assays (individual curves shown in **Figure 4—figure supplement 4C**). Significance is shown for comparisons of each mutant to WT; **p<0.005, ***p<0.001. For (**F**) and (**G**), each bar represents 9 measurements from three independent experiments (mean ± SD).

DOI: https://doi.org/10.7554/eLife.32733.009

The following figure supplements are available for figure 4:

**Figure supplement 1.** Several VWMD mutations reduce the protein level of the affected subunit in cells.

DOI: https://doi.org/10.7554/eLife.32733.010

**Figure supplement 2.** VWMD mutant cells have normal rates of protein synthesis and ISRIB partially rescues a stress-induced reduction.

DOI: https://doi.org/10.7554/eLife.32733.011

**Figure supplement 3.** VWMD mutant cells show a similar response to ER stress as WT cells.

DOI: https://doi.org/10.7554/eLife.32733.012

**Figure supplement 4.** Lysates from VWMD cells exhibit reduced GEF activity in vitro that is enhanced by ISRIB.

DOI: https://doi.org/10.7554/eLife.32733.013

our growth conditions, ISRIB can suppress exogenous stress-induced ATF4 upregulation in cells expressing various eIF2B mutant subunits.

In order to quantitatively compare the response of the different cell lines to ER stress, we took advantage of the genetically encoded ATF4-luciferase reporter. Cells were treated with varying concentrations of thapsigargin (Tg), an inhibitor of the sarco/endoplasmic reticulum $Ca^{2+}$ ATPase, which disrupts ER calcium homeostasis and activates PERK kinase (**Lytton et al., 1991**). After 7 hr of treatment, wild-type cells showed a dose-dependent increase in ATF4-luciferase activity ($EC_{50}$ = 14 nM; **Figure 4—figure supplement 3**). Surprisingly, all VWMD mutant cell lines showed a similar dose response, with no significant differences in Tg $EC_{50}$. This implies that under our assay conditions, VWMD mutations that reduce eIF2B levels (and impair GEF activity in vitro) do not sensitize the cellular response to ER stress. As was the case for Tm, ISRIB prevented activation of the ATF4-luciferase reporter in cells treated with a saturating concentration of Tg, a severe ER stressor (**Figure 4D**).

Finally, we tested whether ISRIB could rescue the reduction in protein synthesis in mutant cells after a stress challenge. We measured $^{35}$S-methionine incorporation following treatment with the ER stressor dithiothreitol (DTT), a reducing agent that causes rapid accumulation of unfolded proteins. After 1 hr of DTT treatment, we observed a reduction in $^{35}$S-methionine incorporation, with similar responses (~75% decrease) in wild-type and VWMD mutant cells (**Figure 4E** and **Figure 4—figure supplement 2D**). Co-treatment with ISRIB significantly restored $^{35}$S-methionine incorporation to a similar extent in both wild-type and VWMD mutant cells. Taken together, our results demonstrate that ISRIB engages VWMD mutant eIF2B in all tested cell lines and attenuates both cellular outputs of the ISR.

## VWMD mutant cell lines have impaired eIF2B GEF activity that is rescued by ISRIB

VWMD mutant HEK293T cells did not exhibit basal ISR activation or increased sensitivity to exogenous ER stress. Given that three of the generated mutations (αV183F, δR483W and εR195H) had a detrimental effect on complex stability and GEF activity of recombinant eIF2B, we speculated that the lack of impairment of VWMD mutant cancer cells was due to high levels of eIF2B expression. Indeed, when we assessed the levels of each eIF2B subunit in a panel of five cell lines (HEK293T,

HeLa, B-lymphocytes, WI-38 and hIPSCs), we discovered that HEK293T cells had the highest expression of every eIF2B subunit (*Figure 4—figure supplement 4A–B*). Our data are consistent with previous reports demonstrating overexpression of eIF2B subunits in transformed mammary epithelial cell lines (*Balachandran and Barber, 2004*; *Kim et al., 2000*), and is likely due to the selective advantage conferred by increased protein synthesis. Given this result, we postulated that conditions wherein the effective concentration of eIF2B is greatly reduced may reveal deficiencies in these mutants.

We produced lysates from wild-type and VWMD mutant cell lines and compared their activities using the GEF assay (*Figure 4F* and *Figure 4—figure supplement 4C*). The >100X dilution of lysates relative to native cytosol placed the system in a regime wherein eIF2B activity became rate-limiting, finally revealing the deleterious nature of each VWMD mutation that was buffered by the high eIF2B levels found in cancer cells. αV183F lysates retained only 32–43% of wild-type activity. Lysates from both γI346T clones possessed 55% of wild-type activity. The observation that one clone had 80% of eIF2Bγ protein compared to wild-type whereas the second had only 10% (*Figure 4A* and *Figure 4—figure supplement 1C*), suggests that the quantity of eIF2Bγ is not limiting for eIF2B complex formation here. The severe disease mutation δR483W retained only 28–35% of wild-type activity. The two mutations in the catalytic subunit, εR113H and εR195H, had 44–51% and 48–63% of wild-type activity, respectively. Importantly, the reductions in activity for the mutants are consistent with data obtained from our fully reconstituted system. Our results represent the first comparison of the effects of VWMD mutations on GEF activity in an isogenic mammalian cell background.

Finally, we tested the ability of ISRIB to rescue the deficiencies of the VWMD lysates in this assay. At 50 nM ISRIB, the $t_{1/2}$ of GDP release with wild-type lysate was 3.6 mins, a 5.5X stimulation of activity (*Figure 4F*). Addition of ISRIB enhanced GEF activity to the point that all reactions containing VWMD lysates attained $t_{1/2}$ < 9 mins. In the case of one of the δR483W clones, this translated to a dramatic 14X improvement in activity. Dose-response analysis showed that ISRIB had an in vitro $EC_{50}$ = 0.38 nM in wild-type lysate (*Figure 4G* and *Figure 4—figure supplement 4C*). Moreover, the $EC_{50}$ values for the VWMD lysates were similar to wild-type, with the notable exception of one αV183F clone ($EC_{50}$ = 0.99 nM). As we observed in our reconstituted system, the two δR483W clones had lower ISRIB $EC_{50}$ than wild-type (0.21 nM and 0.05 nM; see Discussion). This suggests that under the dilute protein concentrations used in this assay, the imbalance in the relative concentrations of the different eIF2B subunits did not dictate ISRIB sensitivity. Taken together, our results demonstrate that VWMD mutant cancer cell lines have impaired GEF activity that is revealed by dilution of the cellular lysate. Moreover, lysate GEF activity can be enhanced significantly, beyond basal wild-type levels, by the small molecule eIF2B activator ISRIB.

## Discussion

### Correlation between eIF2B GEF activity and disease severity of VWMD mutants

Previous work demonstrated that many VWMD mutations lead to a partial reduction in GEF activity. Although the majority of these experiments measured the enzymatic activity of lysates derived from patient cells, a few were performed in yeast cells in which analogous mutations were generated in the highly conserved eIF2B complex (*Richardson et al., 2004*). Intriguingly, some pathogenic mutations have been reported not to affect the enzymatic activity of eIF2B. Liu et al. performed a comprehensive analysis of VWMD mutations using affinity-purified eIF2B derived from HEK293T cells overexpressing all the subunits of the complex, and concluded that GEF activity was not predictive of VWMD severity (*Liu et al., 2011*). Moreover, they reported that some severe mutations affected eIF2B function without obviously impairing recombinant complex integrity or enzymatic activity. By contrast, Fogli et al. measured the GEF activity of transformed lymphocyte lysates derived from 30 VWMD patients and reported a 20–70% decrease in activity in all mutated cells that correlated with age of disease onset (*Fogli et al., 2004b*).

These contrasting conclusions underscore the need for further evaluation of the effect of VWMD mutations on eIF2B function. To this end, we interrogated the effect of pathogenic VWMD mutations in recombinant eIF2B complexes as well as in an isogenic cell model. The two complementary

systems also served as the foundation to evaluate the ability of the eIF2B activator ISRIB to interact with the various VWMD mutant complexes and enhance their residual GEF activity.

In enzymatic assays with both VWMD HEK293T cell lysates and purified recombinant eIF2B complexes, we observed compromised GEF activity in all generated mutants with the exception of εR136H. Importantly, we saw a correlation in GEF activity impairment for each specific mutant using eIF2B from the two different sources. δR483W and αV183F were the most crippling mutations in both assays. γI346T, εR113H and εR195H had similar but milder deleterious effects on GEF activity in cell lysates. In line with lysate measurements, recombinant εR195H eIF2B complexes also showed a milder defect in enzymatic activity than αV183F and δR483W. The $EC_{50}$ of ISRIB for stimulating GEF activity was similar between wild-type and mutant complexes, except for δR483W (in both recombinant protein and cell lysates). Combined with our knowledge that no existing pathogenic mutations map to the putative ISRIB binding site within eIF2B, our data suggest that a broad range of VWMD mutations may be amenable to ISRIB-mediated rescue.

Among our tested mutations, δR483W was the only one within the $(\beta\delta)_2$ regulatory core, and its position suggests that it could have potential effects on the stability of the βδγε subcomplex itself (*Figure 2A*). Although this is not apparent in the μM concentration range of our SEC experiments, they could manifest in the dilute nM regime of the GEF assay. We speculate that the ISRIB $EC_{50}$ for this mutant is lowered because the compound is acting synergistically with eIF2Bα to rescue not just decamer formation but the stability of the tetramer itself. Future experiments using more sensitive methods to assay stability of the βδγε subcomplex at ultra-low concentrations could address this particular question.

The degree of impairment of GEF activity in our in vitro assays and the reported severity of the disease allele in humans did not correlate. In particular, the two catalytic subunit mutants, εR113H (which has a classical disease presentation) and εR195H (which presents with a severe early phenotype), had similar lysate GEF activity. Thus, our reconstituted system can detect defects in eIF2B enzymatic function but does not capture true cellular complexity, such as factors that may modulate the activity of eIF2B in astrocytes or oligodendrocytes – the cell types most severely affected in the disease.

Recombinant eIF2B complexes of the εR136H mutant had wild-type levels of activity and complex stability. We were unable to generate HEK293T cells carrying this mutation and thus could not measure the enzymatic activity of cell lysates. The GEF activity of cerebellar lysates prepared from mice carrying the analogous mutation showed a 23% reduction in activity, and no significant decrease in the level of the mutated subunit was observed (*Geva et al., 2010*). In this case, the purified mutant complex did not reveal the enzymatic defect that is observed in the relevant cell types. eIF2B has been reported to possess non-GEF activities, functioning as a guanine nucleotide dissociation inhibitor (GDI) displacement factor (*Jennings et al., 2013*), as well as directly regulating ternary complex formation (*Jennings et al., 2017*). Mutations that affect these functions will not be revealed in the standard GEF assay that measures release of GDP from pre-loaded recombinant eIF2. It is worth noting that the αβδ regulatory subcomplex is evolutionarily descended from an archaeal sugar-binding metabolic enzyme complex, ribose 1,5-biphosphate isomerase, and has been shown to retain its capacity to bind metabolites (*Dholakia et al., 1986*; *Gross et al., 1988*; *Kuhle et al., 2015*). This suggests that the absence or presence of metabolites could act as another layer of regulation on eIF2B function. VWMD mutations affecting any of these activities/regulators would not be detected in the minimal system represented by our GEF assay. We note that this class of mutations provide an ideal opportunity to further dissect eIF2B function in future experiments, without the complication of intrinsically impaired GEF activity.

## The eIF2B activator ISRIB promotes decamer formation of VWMD mutant eIF2B complexes and rescues GEF activity

Mutations that destabilize the eIF2B decamer compromise GEF activity and may potentiate the disease phenotype. ISRIB binds at the eIF2B two-fold symmetric axis formed by $(\beta\delta)_2$, stabilizing and activating the complex. As eIF2B is essential for life, VWMD mutations can only partially cripple its activity. We hypothesized that regardless of the VWMD mutations introduced, ISRIB would be able to interact with the mutant complexes and enhance the residual GEF activity. Our data demonstrate that ISRIB can indeed interact with eIF2B complexes harboring pathogenic VWMD mutations. In cells, the activator can fully restore GEF activity and suppress induction of the ISR following ER stress

and restore protein synthesis in five different mutant cell lines. Notably, ISRIB rescued decamer formation of all recombinant VWMD mutant eIF2B complexes that displayed a significant reduction in their decamer:tetramer ratio. This stabilization of mutant eIF2B by ISRIB resulted in restoration of GEF activity against its substrate eIF2 to wild-type levels. As predicted, incorporation of an ISRIB-resistant mutation, δL179F, prevented activation of a complex that contains the destabilizing αV183F mutation.

The observation that ISRIB enhanced the activity of wild-type eIF2B in our cell-based assays suggests that even the wild-type complex can benefit from additional stabilization. We do not rule out the intriguing possibility that ISRIB may also have an allosteric stimulatory effect on the eIF2B complex, which is implied by our finding that ISRIB reduces the sensitivity of eIF2B to inhibition by phospho-eIF2 (*Figure 1G*). This result diverges from our previous observation that ISRIB does not alter the response of eIF2B to phospho-eIF2 (*Sidrauski et al., 2015*). However, our previous results were obtained using a high phospho-eIF2 concentration (~175 nM compared to 0–50 nM here) and lower eIF2B concentration (~0.5 nM compared to 1 nM here), conditions under which ISRIB is unlikely to have a rescuing effect.

The effect of ISRIB on white matter astrocytes and oligodendrocytes, the most susceptible cell types in VWMD, must be investigated. Further experiments are also needed to understand the differential sensitivity of different cell types to VWMD mutations. Post-translational modifications and metabolites may modify eIF2B stability and activity in cells and these could vary significantly depending on cellular context. The intracellular concentration of eIF2B subcomplexes as well as the ratio of eIF2:eIF2B are likely also important, particularly since eIF2 can convert from a substrate to an inhibitor of eIF2B upon stress-induced phosphorylation.

In conclusion, the ability of ISRIB to restore the GEF activity of eIF2B complexes containing VWMD mutations suggests that it may have beneficial effects in this neurodevelopmental leukodystrophy. More broadly, the finding that ISRIB can attenuate ISR activation in cells opens up avenues for use of this compound to ameliorate conditions wherein this crucial signaling pathway is inappropriately engaged.

## Materials and methods

**Key resources table**

| Reagent type or resource | Designation | Source or reference | Identifiers | Additional information |
|---|---|---|---|---|
| Cell line (*H. sapiens*) | HEK293T with ATF4-Luc reporter | PMID: 23741617 | | |
| Cell line (*H. sapiens*) | HEK293T (eIF2Bα V183F) with ATF4-Luc reporter | This paper | | two clones generated |
| Cell line (*H. sapiens*) | HEK293T (eIF2Bγ I346T) with ATF4-Luc reporter | This paper | | two clones generated |
| Cell line (*H. sapiens*) | HEK293T (eIF2Bδ R483W) with ATF4-Luc reporter | This paper | | two clones generated |
| Cell line (*H. sapiens*) | HEK293T (eIF2Bε R113H) with ATF4-Luc reporter | This paper | | two clones generated |
| Cell line (*H. sapiens*) | HEK293T (eIF2Bε R195H) with ATF4-Luc reporter | This paper | | two clones generated |
| Antibody | Rabbit monoclonal anti-ATF4 | Cell Signaling | #11815 | (1:50) in Wes |
| Antibody | Rabbit monoclonal anti-eIF2α | Cell Signaling | #5324 | (1:100) in Wes |
| Antibody | Rabbit polyclonal anti-eIF2Bα | ProteinTech | #18010–1-AP | (1:50) in Wes |
| Antibody | Rabbit polyclonal anti-eIF2Bβ | ProteinTech | #11034–1-AP | (1:50) in Wes |
| Antibody | Rabbit polyclonal anti-eIF2Bγ | ProteinTech | #11296–2-AP | (1:25) in Wes |
| Antibody | Rabbit polyclonal anti-eIF2Bδ | ProteinTech | #11332–1-AP | (1:50) in Wes |
| Antibody | Rabbit polyclonal anti-eIF2Bε | Bethyl Labs | #A302-556 | (1:50) in Wes |
| Antibody | Mouse monoclonal anti-tubulin | Cell Signaling | #3873 | (1:50) in Wes |

*Continued on next page*

*Continued*

| Reagent type or resource | Designation | Source or reference | Identifiers | Additional information |
|---|---|---|---|---|
| Recombinant DNA reagent | CRISPR nuclease vector with OFP reporter | Thermo Fisher | #A21174 | |
| Sequence-based reagent | eIF2Bα V183F guide RNA | This paper | | GTGGTGCTAGATGCTGCTGTCGG |
| Sequence-based reagent | eIF2Bγ I346T guide RNA | This paper | | TGACAATCTGGGCTGACGAATGG |
| Sequence-based reagent | eIF2Bδ R483W guide RNA | This paper | | GACTAGATTCAACAACCGTAGGG |
| Sequence-based reagent | eIF2Bε R113H guide RNA | This paper | | CCGCCCTACATCTCTCAATGTGG |
| Sequence-based reagent | eIF2Bε R195H guide RNA | This paper | | TTGTCTTCGTGGCAACGAGTTGG |
| Recombinant protein | GST-PERK | Thermo Fisher | #PV5106 | Used to phosphorylate eIF2 in vitro |
| Commercial assay | ONE-GLO luciferase assay | Promega | #E6120 | |
| Chemical compound | Bodipy-FL-GDP | Thermo Fisher | #G22360 | |
| Chemical compound | ISRIB | PMID: 23741617 | | Synthesized in-house |
| Chemical compound | Thapsigargin | Sigma-Aldrich | #T9033 | Stock solution prepared in DMSO |
| Chemical compound | Tunicamycin | Sigma-Aldrich | #T7765 | Stock solution prepared in DMSO |

## Cell lines

The HEK293T ATF4-luciferase reporter cell line was previously described (*Sidrauski et al., 2013*). A monoclonal population (isolated by single cell dilution) was used as the parental cell line for generating the VWMD knock-in mutations. Cells were maintained in DMEM High Glucose (Corning CellGro) supplemented with 10% FBS (Gibco) and 1X antibiotic-antimycotic solution (Gibco).

eIF2B mutations were introduced into the HEK293T ATF4-luciferase reporter cell line using the GeneArt CRISPR Nuclease Vector with OFP Reporter Kit (Thermo Fisher). gRNA sequences (see Key Resources Table) were designed with MIT's online CRISPR design tool (crispr.mit.edu), synthesized by IDT and inserted into the GeneArt plasmid according to manufacturer's instructions. 150-base oligonucleotides were synthesized and used as DNA donor templates for homology-directed repair (HDR) after Cas9 cutting. The oligonucleotides were designed to include the point mutation of interest, as well as silent mutations to the gRNA PAM sequence to prevent Cas9 re-cutting.

1 μg of the guide-containing GeneArt plasmid and 10 μL of 50 μM HDR donor template were delivered into cells with a Lonza 4D-Nucleofector, using the SF reagent kit and program SF-130, according to vendor's instructions. In some cases, cells were temporarily arrested at the G2/M transition by treatment with 20 ng/mL nocozadole for 16 hr before nucleofection to improve HDR efficiency. After 48–72 hr, OFP-positive cells were single-sorted into 96-well plates using a BD BioSciences FACSAria Fusion. Single clones were expanded and genomic DNA was extracted with a Quick-gDNA MiniPrep kit (Zymo Research). The eIF2B locus of interest was amplified by PCR and the purified PCR product was subjected to Sanger sequencing. Due to the low efficiency of HDR, this step typically yielded mixed sequences due to differential editing of the three eIF2B alleles in each clone. To disambiguate the editing events, PCR products were cloned into TOPO vectors using the TOPO-TA cloning kit (Thermo Fisher, Waltham, MA). TOPO clones were sequenced to ensure that we determined the identity of each of the three alleles (*Supplementary file 1*).

Cell line identities were verified by ATCC STR profiling and have been tested to be free of mycoplasma contamination.

## ATF4-luciferase reporter assay

One day prior to treatment, cells were seeded into 96-well white opaque plates (Falcon) at a density of $4 \times 10^4$ cells/well in 100 μL medium/well. Thapsigargin (1 mM stock concentration in DMSO) was dispensed into wells using a Tecan D300e digital dispenser, and incubated at 37°C for the specified time. ONE-Glo luciferase assay reagent (Promega) was added to each well following manufacturer's

instructions and incubated for 5 min at room temperature. Luminescence was measured on a Molecular Devices SpectraMax i3x plate reader.

## $^{35}$S-methionine incorporation assays

One day prior to the assay, cells were seeded into 12-well tissue culture plates coated with 10 mg/mL poly-D-lysine (Sigma) at a density of $5 \times 10^6$ cells/mL. On the day of the assay, cells were treated with the indicated concentrations of DTT and ISRIB, and incubated for 40 min. 20 µCi of EasyTag L-[$^{35}$S]-methionine (Perkin Elmer) was added to each well and the plate was incubated for an additional 20 min. Cells were washed with PBS and lysed by addition of 0.2 mL RIPA buffer supplemented with protease and phosphatase inhibitor cocktail (Pierce).

For measurement of $^{35}$S-methionine incorporation by gel quantitation, 20 µL of each lysate was run on a 10% SDS-PAGE gel (Bio-Rad). Gels were fixed in 40% methanol/10% acetic acid, stained with Bio-Rad QC Colloidal Coomassie solution, rinsed and dried. Dried gels were exposed to a storage phosphor screen (GE Healthcare) for 1–2 days. The phosphor screen was imaged on a Typhoon FLA scanner (GE Healthcare). For each lane of the phosphor and Coomassie images, areas and mean pixel intensities were quantified using ImageJ (NIH.gov).

## Protein purification

Sequences of full-length human eIF2B subunits (UniProt accession numbers: eIF2Bα, Q14232; eIF2Bβ, P49770; eIF2Bγ, Q9UI10; eIF2Bδ, Q9NR50; eIF2Bε, Q13144) and human eIF2 subunits (UniProt accession numbers: eIF2S1, P05198; eIF2S2, P20042; eIF2S3, P41091) were synthesized by GenScript with or without an N-terminal FLAG-TEV tag, and cloned into a modified pHybE vector for expression (*Hsieh, 2012*). Point mutations were introduced into subunits by site-directed mutagenesis.

Proteins were expressed in HEK293-6E cells (NRC-Canada) using a Wave Bioreactor System 20/50EHT (GE Healthcare). Cells were grown to a density of $1.2 \times 10^6$ cells/mL in Freestyle 293 expression media (Thermo Fisher) supplemented with 0.05% Pluronic-F68 (Thermo Fisher) and 0.5% Penicillin-Streptomycin (Thermo Fisher). Cells were transfected with expression plasmid DNA at 0.5 mg/L mixed with 1 mg/mL polyethylenimine (PolySciences Inc.) in a 1:4 ratio for 10 min, and grown at 37°C with 8% $CO_2$ for 3 days.

All purification steps were performed at 4°C. Cell pellets were resuspended in buffer A (20 mM HEPES, 300 mM KCl, pH 7.4) and lysed in a French Press (Aminco). Post-lysis, 1 mM EDTA and a standard protease inhibitor cocktail that omitted AEBSF but included PMSF were added to the supernatant. Protein complexes were purified by FLAG affinity chromatography (*Domanski et al., 2012*) using buffer A + 1 mM EDTA. Pooled fractions were dialyzed overnight against buffer A + 1 mM EDTA and 0.5 mM TCEP. eIF2 was further purified on a TSKgel SuperQ-5PW anion-exchange column (50 mL bed volume; Tosoh Biosciences) using 50 mM HEPES, 1 mM MgCl$_2$, 50–500 mM KCl salt gradient, 8.0–7.2 pH gradient. Protein concentrations were determined by Bradford assay.

For verification of protein complex sizes by multi-angle light scattering, 100 µL of each undiluted purified sample (concentrations ranged from 0.55 to 0.75 mg/mL) was loaded onto a WTC-030S5 column (Wyatt Technology) by auto-injection from a 1260 Infinity II HPLC system (Agilent Technologies). The column was connected in-line to a Dawn HELIOS II system (Wyatt Technology). The system was run at 0.3 mL/min for 1 hr using 20 mM HEPES, 200 mM KCl, 5 mM MgCl$_2$, 1 mM TCEP, pH 7.4 as the mobile phase. Protein concentrations for molecular weight calculations were determined by UV$_{280}$ absorbance.

## eIF2 loading with fluorescent nucleotide and phosphorylation

For loading of eIF2 with fluorescent nucleotide, 500 µL of eIF2 at 4.9 mg/mL (38 µM) was adjusted to a final concentration of 5 mM EDTA. Bodipy-FL-GDP (Thermo Fisher #G22360) was added to a 5-fold molar excess over eIF2 and the mixture was incubated for 1 hr at room temperature. MgCl$_2$ was added to a final concentration of 10 mM to quench the loading reaction. The mixture was buffer-exchanged by loading onto an Illustra NAP-5 column (GE Healthcare) equilibrated with 20 mM HEPES, 120 mM KCl, 10 mM MgCl$_2$, 1 mM TCEP, pH 7.4. Loaded eIF2 was eluted with 1 mL buffer, aliquoted and stored at −80°C until use.

For phosphorylation of eIF2, 150 µL of eIF2 at 1.5 µM was incubated with 500 µM ATP and 100 nM PERK kinase (Thermo Fisher #PV5106) for 2 hr at room temperature. The mixture was buffer-exchanged by loading onto a PD Spintrap G-25 column (GE Healthcare) equilibrated with 20 mM HEPES, 120 mM KCl, 5 mM MgCl$_2$, 1 mM TCEP, pH 7.4. Phosphorylated eIF2 was eluted with 0.6 mL buffer, aliquoted and stored at −80°C until use.

### HEK293T lysates

Wild-type and VWMD HEK293T cells were grown to approximately 80% confluence in 4 × 15 cm tissue culture dishes per cell line. Cells were harvested in PBS by pipetting and centrifuged to pellet. Each cell pellet was resuspended in 0.8 mL 20 mM HEPES, 150 mM KCl, 2 mM TCEP, pH 7.4 + cOmplete$^{TM}$ EDTA-Free protease inhibitor cocktail (Roche). Cells were lysed by manually passing them through a tungsten carbide ball-bearing homogenizer (isobiotec) with an 8 µm clearance. Lysed samples were cleared by centrifugation at 48,000 x $g$ for 15 min at 4°C. Cleared lysates were aliquoted and stored at −80°C until use.

### Western blots

Wild-type and VWMD HEK293T lysates were normalized to a protein concentration of 0.2 mg/mL by BCA assay. Samples were run on a ProteinSimple Wes capillary system using a 12–230 kDa separation module and the following instrument parameters: run conditions = 375V, 25 min; antibody diluent time = 5 min; primary and secondary antibody incubation time = 30 min. All antibodies were used at a dilution of 1:50 except for eIF2Bγ (1:25 dilution) and eIF2α (1:100 dilution).

### GEF assay

Bodipy-FL-GDP-loaded eIF2 was adjusted to 1 µM using assay buffer (20 mM HEPES, 120 mM KCl, 5 mM MgCl$_2$, 1 mM TCEP, pH 7.4). A mixture of 50 nM eIF2 +1 mg/mL BSA was prepared in assay buffer and loaded into the injection module of a Molecular Devices SpectraMax i3x plate reader.

2X mixtures were prepared in assay buffer for the following experiments: HEK293T lysate assay = 0.2 mM GDP, 1 mg/mL BSA, 10 nM phospho-eIF2, 0.6 mg/mL lysate; recombinant eIF2B assay = 0.2 mM GDP, 1 mg/mL BSA, 10 nM phospho-eIF2, 6 nM eIF2B 4mer + α subunit (or 2 nM eIF2B 5mer); α titration assay = 0.2 mM GDP, 1 mg/mL BSA, 10 nM phospho-eIF2, 10 nM eIF2B 4mer, 0–1 µM α subunit; phospho-eIF2 titration assay = 0.2 mM GDP, 1 mg/mL BSA, 2 nM eIF2B 5mer, phospho-eIF2 and 'cold' eIF2 in varying ratios with a total concentration of 100 nM. In all experiments, 5 µL of each mixture was pipetted into the wells of a 384-well black low-volume plate (Greiner Bio-One #784076).

For ISRIB dose-response studies, ISRIB was dispensed from a 25 µM or 250 µM stock using a Tecan D300e digital dispenser into wells containing the mixtures above. DMSO was used for normalization in order to maintain a concentration of 0.2% in all wells in the final assay volume. For each run, triplicate measurements were made for each concentration of ISRIB.

Reactions were read on the Spectramax plate reader using the following instrument parameters: plate temperature = 25°C; eIF2 injection = 5 µL per well; shake before read = 2 s low intensity; PMT sensitivity = high; flashes/read = 6; excitation wavelength = 485 nm (15 nm width); emission wavelength = 535 nm (25 nm width); read duration = 30 min at 45 s intervals.

In a final assay volume of 10 µL/well, the following conditions were kept constant: 25 nM Bodipy-FL-GDP-loaded eIF2, 0.1 mM GDP, 1 mg/mL BSA. Phospho-eIF2, ISRIB and eIF2B varied depending on the experiment. To calculate GDP release half-lives, single-exponential decays were fit to the raw fluorescence intensity data using Prism (GraphPad Software).

### Size-exclusion chromatography

Purified proteins were thawed and centrifuged at 20,000 x $g$ for 10 min at 4°C to remove any potential aggregates. For 5mer runs, proteins were normalized to 1 µM prior to column loading. For 4mer + α runs, the 4mer was normalized to 1 µM and the α subunit was normalized to 1.2 µM. 100 µL of each sample was injected onto a Superose 6 10/300 column connected to an AKTA Pure 25 FPLC system (both from GE Healthcare). The system was run at 0.4 mL/min for 1 hr using 20 mM HEPES, 300 mM NaCl, 2 mM MgCl$_2$, 1 mM TCEP, pH 7.4 as the mobile phase. For conditions with ISRIB, the protein samples and mobile phase were supplemented with 200 nM ISRIB.

## Data processing

Data was processed in Prism (GraphPad Software). One- or two-way ANOVA with Fisher's LSD was used for significance testing between different samples.

## Acknowledgements

The authors thank John Wang and Dan Eaton for assistance with multi-angle light scattering measurements. We also thank Swathi Krishnan, Calvin Jan, Voytek Okreglak, Jeff Settleman, David Stokoe and David Botstein for critical reading of the manuscript.

## Additional information

### Competing interests

Yao Liang Wong and Lauren LeBon: The authors were employees of Calico Life Sciences LLC at the time the study was conducted and have no other competing financial interests to declare. Rohinton Edalji, Hock Ben Lim, Chaohong Sun: The authors were employees of AbbVie at the time the study was conducted and have no other competing financial interests to declare. Carmela Sidrauski: The author was an employee of Calico Life Sciences LLC at the time the study was conducted. The author is listed as an inventor on a patent application describing ISRIB. Rights to the patent application have been licensed to Calico Life Sciences LLC from the University of California, San Francisco. The author has no other competing financial interests to declare.

### Funding

| Funder | Author |
| --- | --- |
| Calico Life Sciences LLC | Yao Liang Wong<br>Lauren LeBon<br>Rohinton Edalji<br>Hock Ben Lim<br>Chaohong Sun<br>Carmela Sidrauski |

This study was funded by Calico Life Sciences LLC.

### Author contributions

Yao Liang Wong, Lauren LeBon, Conceptualization, Resources, Data curation, Formal analysis, Validation, Investigation, Visualization, Methodology, Writing—original draft, Writing—review and editing; Rohinton Edalji, Resources, Formal analysis, Investigation, Methodology, Writing—review and editing; Hock Ben Lim, Investigation, Methodology; Chaohong Sun, Resources, Supervision, Methodology, Writing—review and editing; Carmela Sidrauski, Conceptualization, Resources, Supervision, Funding acquisition, Methodology, Writing—original draft, Writing—review and editing

### Author ORCIDs

Yao Liang Wong ⓘD http://orcid.org/0000-0003-0298-8510
Lauren LeBon ⓘD https://orcid.org/0000-0003-3205-1948
Carmela Sidrauski ⓘD https://orcid.org/0000-0002-4850-3112

### Decision letter and Author response

Decision letter https://doi.org/10.7554/eLife.32733.018
Author response https://doi.org/10.7554/eLife.32733.019

## Additional files

### Supplementary files

• Supplementary file 1. Allele sequences of HEK293T mutant cell lines used in this study.
DOI: https://doi.org/10.7554/eLife.32733.014

• Transparent reporting form
DOI: https://doi.org/10.7554/eLife.32733.015

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
