## [Decision Letter]

Thank you for submitting your article "The small molecule ISRIB rescues the stability and activity of Vanishing White Matter Disease eIF2B mutant complexes" for consideration by *eLife*. Your article has been favorably evaluated by Randy Schekman (Senior Editor) and three reviewers, one of whom, Alan Hinnebusch, is a member of our Board of Reviewing Editors. The reviewers have opted to remain anonymous.

The reviewers have discussed the reviews with one another and the Reviewing Editor has drafted this decision to help you prepare a revised submission.

Summary:

In this study, the authors have reconstituted WT and VWM eIF2B heterodecameric complexes from purified tetramers and α-α homodimers, and shown that 3 of 4 VWM substitutions examined reduce GEF activity and can be rescued to WT basal activity (i.e. in absence of ISRIB) by addition of ISRIB. They also show that the α-α homodimer can stimulate the GEF activity of the eIF2B tetramer formed with the other 4 eIF2B subunits. In the case of the α-subunit mutation V183F, located at the α-α interface, elevated concentrations of the mutant subunit were required to observe its stimulatory effect on the GEF activity of the tetramer, and at a concentration of this α variant where half-maximal stimulation of GEF activity was achieved, ISRIB could stimulate the activity further to achieve the WT basal level. Using gel filtration (SEC), they then showed that 3 of the 4 mutants impair decamer formation at an elevated salt concentration chosen to slightly destabilize the WT decamer, although the effect of R195H is marginal; and demonstrated that ISRIB can rescue stable heteropentamer formation for all 3 variants, in parallel with rescuing their GEF activities. They went on to examine a site-directed substitution in the δ subunit, L487W, predicted to perturb the δ-β interface and also the predicted ISRIB binding pocket, finding that it destabilizes the heteropentamer and reduces GEF activity in a manner that cannot be rescued by ISRIB. These findings provide direct evidence that heteropentamer formation is required for robust eIF2B activity, and that ISRIB enhances GEF activity by restoring heteropentamer formation. Consistent with these findings, they also showed that combining the α-subunit V183F VWM substitution with a previously described δL179F substitution, which confers resistance to the stimulatory effect of ISRIB without affecting GEF activity, yields a reconstituted double mutant complex with greatly impaired activity that cannot be rescued by ISRIB at all. The authors went on to use gene-editing to generate isogenic cell lines derived from transformed HEK293T cells containing five different VWM mutations as the only source of the relevant eIF2B subunit; some of which reduced the levels of the mutated subunit, or one or more other eIF2B subunits. Surprisingly, despite the strong effects of V183F and R483W on heteropentamer formation and eIF2B activity in vitro, neither of these substitutions, nor any of the other 3 tested, induced ATF4 expression or reduce 35S-Met incorporation in non-stressed cells. Even in response to tunicamycin or Thapsigargin (Tg), which activate PERK, the VWM substitutions did not increase induction of endogenous ATF4 protein. And using the ATF4-LUC reporter, none of the mutations conferred increased reporter expression at Tg concentrations below the EC^50^ for reporter induction in the WT cells. Similar results were obtained using two different concentrations of DTT and measuring 35S-Met incorporation. Thus, the VWM mutations do not sensitize the cellular response to ER stress, which the authors attribute to elevated eIF2B levels in these transformed cells. However, for all mutants tested, ISRIB can suppress the induction of ATF4 protein, ATF4-LUC expression and reduced 35S-Met incorporation. They were able to recapitulate the deleterious effects of the three VWM substitutions also examined with recombinant complexes, and two others, on GEF activity when the cell lysates employed in the assays were highly diluted, and again, the reductions in GEF activity were completely rescued by ISRIB. The two VWM substitutions that reduced GEF activity the most using recombinant complexes (V183F and R483W), also reduced GEF activity in the cell lysates more so than the third (R195H). However, considering the results for all five mutants examined in lysates, they find that the extents of the GEF defects do not correlate with the severity of the disease, at least when equating age of onset with disease severity.

Essential revisions:

1) As a follow up to Figure 1, document the extent of holocomplex formation achieved on mixing WT tetramers and α2 dimers.

2) Provide a rigorous statistical analysis of differences in GEF activity and holocomplex assembly for the different mutant complexes and adjust conclusions accordingly.

3) Conduct experiments to determine whether ISRIB increases the GEF activity of the tetramer by enhancing octamer formation.

4) Resolve discrepancies between the two panels of Figure 2 regarding GEF activity of the tetramer, clearly shown on the left panel but seemingly contradicted by the right panel.

5) Improve Figure 2 by showing western blots of fractions from the size exclusion chromatography experiments.

6) Quantify the ATF4 protein expression for the experiments shown in Figure 4/Figure 4—figure supplement 2, and discuss any differences that might exist among the mutants.

7) Examine the GEF activity of the recombinant R113H mutant complex, which has been analyzed in all previous studies on VWM mutant complexes.

8) Measure eIF2B expression levels in HEK cells to determine if they truly are elevated compared to untransformed cells.

9) Discuss the physiological relevance of using high salt to measure effects of VWM mutations on complex formation; and possibly refrain from inferences that the in vivo defects conferred by V183F or R483W arise from weaker heteropentamer assembly.

10) Re-think the interpretation of the lower ISRIB EC50 for the R483W variant, which doesn't seem to make sense.

*Reviewer #1:*

In this study, the authors have reconstituted WT and VWM eIF2B heterodecameric complexes from purified tetramers and α-α homodimers, and shown that 3 of 4 VWM substitutions examined reduce GEF activity and can be rescued to WT basal activity (i.e. in absence of ISRIB) by addition of ISRIB. They also show that the α-α homodimer can stimulate the GEF activity of the eIF2B tetramer formed with the other 4 eIF2B subunits. In the case of the α-subunit mutation V183F, located at the α-α interface, elevated concentrations of the mutant subunit were required to observe its stimulatory effect on the GEF activity of the tetramer, and at a concentration of this α variant where half-maximal stimulation of GEF activity was achieved, ISRIB could stimulate the activity further to achieve the WT basal level. Using gel filtration (SEC), they then showed that 3 of the 4 mutants impair decamer formation at an elevated salt concentration chosen to slightly destabilize the WT decamer, although the effect of R195H is marginal; and demonstrated that ISRIB can rescue stable heteropentamer formation for all 3 variants, in parallel with rescuing their GEF activities. They went on to examine a site-directed substitution in the δ subunit, L487W, predicted to perturb the δ-β interface and also the predicted ISRIB binding pocket, finding that it destabilizes the heteropentamer and reduces GEF activity in a manner that cannot be rescued by ISRIB. These findings provide direct evidence that heteropentamer formation is required for robust eIF2B activity, and that ISRIB enhances GEF activity by restoring heteropentamer formation. Consistent with these findings, they also showed that combining the α-subunit V183F VWM substitution with a previously described δL179F substitution, which confers resistance to the stimulatory effect of ISRIB without affecting GEF activity, yields a reconstituted double mutant complex with greatly impaired activity that cannot be rescued by ISRIB at all. The authors went on to use gene-editing to generate isogenic cell lines derived from transformed HEK293T cells containing five different VWM mutations as the only source of the relevant eIF2B subunit; some of which reduced the levels of the mutated subunit, or one or more other eIF2B subunits. Surprisingly, despite the strong effects of V183F and R483W on heteropentamer formation and eIF2B activity in vitro, neither of these substitutions, nor any of the other 3 tested, induced ATF4 expression or reduce 35S-Met incorporation in non-stressed cells. Even in response to tunicamycin or Thapsigargin (Tg), which activate PERK, the VWM substitutions did not increase induction of endogenous ATF4 protein. And using the ATF4-LUC reporter, none of the mutations conferred increased reporter expression at Tg concentrations below the EC_50_ for reporter induction in the WT cells. Similar results were obtained using two different concentrations of DTT and measuring 35S-Met incorporation. Thus, the VWM mutations do not sensitize the cellular response to ER stress, which the authors attribute to elevated eIF2B levels in these transformed cells. However, for all mutants tested, ISRIB can suppress the induction of ATF4 protein, ATF4-LUC expression and reduced 35S-Met incorporation. They were able to recapitulate the deleterious effects of the three VWM substitutions also examined with recombinant complexes, and two others, on GEF activity when the cell lysates employed in the assays were highly diluted, and again, the reductions in GEF activity were completely rescued by ISRIB. The two VWM substitutions that reduced GEF activity the most using recombinant complexes (V183F and R483W), also reduced GEF activity in the cell lysates more so than the third (R195H). However, considering the results for all five mutants examined in lysates, they find that the extents of the GEF defects do not correlate with the severity of the disease, at least when equating age of onset with disease severity.

General critique:

The results are significant in providing direct evidence that heteropentamer formation is critical for robust eIF2B activity in vitro, and underlies the stimulatory effect of the α-α homodimer on GEF activity of the tetramer formed by the other four eIF2B subunits; and that ISRIB rescues GEF activity by promoting heteropentamer formation. The results are also significant in showing that the deleterious effects of 6 different VWM mutations on GEF activity can be rescued to WT levels by ISRIB. The findings that the age of disease onset does not correlate with the severity of the defect in GEF activity for either recombinant complexes or eIF2B complexes in cell lysates represents a rigorous set of results that are germane to this controversial aspect of VWM. Their finding that none of the VWM substitutions generated in HEK cells by gene editing induce the ISR in the absence of stress, nor exacerbate it when PERK is activated, was surprising and might reflect the abnormally high eIF2B levels in the transformed HEK cells. While this is a disappointing aspect of the study, it does not prevent them from showing that ISRIB can rescue eIF2B activity in cell lysates for all 5 VWM mutants tested.

1) MALS analysis should be done on the mixture of WT 4-mer and α2 dimers to show the yield of 10-mer achieved in the reconstitutions analyzed in Figure 1.

2) Subsection “Correlation between eIF2B GEF activity and disease severity of VWMD mutants”, third paragraph, the claim that R483W is more crippling than V183F in assays of both recombinant eIF2B and in cell lysates is not compelling, as there is no demonstration that these differences are statistically significant. It's also unclear whether the activity of R113H in lysates is significantly higher than that of V183F because the variance between the two V183F replicate cell lines is quite large. These interpretations may have to be withdrawn.

3) The statement that "A key finding of our work is that the catalytic activity of recombinant eIF2B is proportional to the stability of the decameric holoenzyme." is an overstatement, as L487W has a smaller effect on decamer stability but a greater effect on activity vs. V183F. What they can say is that several mutations with strong effects on decamer stability also have strong effects on activity, and both defects are co-rescued by ISRIB.

*Reviewer #2:*

This manuscript reports the investigation of the effects of a compound ISRIB on eIF2B activity and regulation. It confirms prior studies that show that ISRIB can stimulate eIF2B activity and promote/stabilize decamer formation. It extends these findings to examine the impact of CACH/VWM disease mutations on eIF2B activities using purified proteins and in CRISPR/cas9 engineered HEK293T cells. ISRIB enhances the activity of all eIF2B mutants tested, except for a mutation in the predicted ISRIB-binding pocket.

The authors propose that ISRIB stabilises decameric eIF2B that by an unknown mechanism boosts its GEF function and this occurs with all VWM mutations tested here. As ISRIB was previously shown to cross the blood-brain barrier it has potential to correct eIF2B function in glial cells sensitive to eIF2B mutations, but this was not addressed in this study.

In general the work appears to have been performed to a high standard, however there are some issues relating to how the experiments are described and presented that should be addressed. In general this study agrees with many prior analyses of VWM alleles in that studies of purified mutant eIF2B can reveal defects, when some cell-based assays do not, and vice versa. The main novel features of this study are the use of CRISPR to make cells lines and the evaluation of ISRIB as a potential curative agent. The conclusions could be strengthened by some additional experiments, which the authors have resources in place to complete.

1) Figure 2. Left panel. This figure appears to show that in the absence of eIF2Bα, ISRIB can still stimulate 4-mer GEF activity. Reactions without eIF2Bα are not shown, but were presumably done? The lowest point is 0.01M eIF2Bα, meaning that almost all eIF2B lacks eIF2Bα, but its activity is still boosted by ISRIB. This is not discussed in the manuscript.

How does ISRIB boost 4-mer activity? Does it stabilize octamer formation in the absence of eIF2Bα? Or does ISRIB boost GEF activity via a different mechanism? Performing size exclusion chromatography of the 4mer alone +/- ISRIB would allow evaluation of the first mechanism. It was shown previously that, unlike the mammalian complexes studied here, the yeast eIF2B complex octamer lacking eIF2Bα is stable (see Gordiyenko et al. PMID:24852487).

2) Figure 2. Similarly there is evidence that both yeast and mammalian eIF2B complexes lacking eIF2Bα are less sensitive to eIF2 phosphorylation, (e.g. Elsby et al. PMID:21795329; Kimball et al. PMID:9582312 for mammalian references). By performing an experiment analogous to the one shown in Figure 1, with 4mer eIF2B +/- ISRIB and V183F +/- ISRIB, the impact of loss of eIF2Bα on sensitivity to the ISR and ISRIB could be assessed. This would enable a comparison of the effects of phospho-eIF2 between purified V183F which has a significant 4mer form (Figure 2) and the 293 cell data in Figure 4.

3) Figure 2 right panel. This presentation is not clear. The plots appear to intersect through 0,0 where the axes meet. This would indicate that the 4mer protein is inactive, which it is not. It be simply that the graph title and legend require changing to indicate what is actually being shown here.

4) Figure 2/3A would be improved by showing example western blots of fractions from the size exclusion chromatography. This should demonstrate that eIF2Bα is absent from the 4mer fraction and may indicate what is present in the low molecular weight fractions. The panels could be presented as a figure supplement.

5) Figure 3 and discussion in subsection “Correlation between eIF2B GEF activity and disease severity of VWMD mutants”. If the authors have the ability to generate labelled ISRIB, they could determine if its affinity for eIF2B is altered by L487W as they suggest, or is altered for the 4mer vs. 10mer forms.

6) Figure 4/Figure 4—figure supplement 2 and text p8 paras 3/4. Figure 4—figure supplement 2 shows that some mutants do have higher ATF4 levels. This is not indicated in the text.

7) Figure 4. This figure does not show data for untreated cells. From the data shown in Figure 4—figure supplement 3, it appears the presentation of 4D is over-normalised and therefore does not show clearly where there is any differences in the eIF2B mutant responses to Tg treatment, only responses to ISRIB. By presenting the data differently more information would be clearer.

*Reviewer #3:*

1) As noted by the authors, some studies suggest that there might be a correlation between eIF2B GEF activity and age of onset in VWMD (e.g. the Fogli and Horzinski references in the present manuscript) whereas other found no association (e.g. the Liu and van Kollenburg references). Based on these discrepancies the authors logically say that further work is needed to clarify the relationship between GEF activity and VWMD. It is therefore surprising that, other than the δR483W mutant, they did not assess the activity of other mutants assessed in previous studies. For example, the four previous studies that are referenced in this paper all showed that the activity of the εR113H mutant was lower than the wild type protein. In the present study, the assembly of that mutant into the holocomplex was examined, but not its GEF activity. The previous studies showed that other homozygous mutants, e.g. εP243L and βE213G, had lower GEF activity compared to wild type. The authors should confirm that their assay replicates the findings previously reported for at least one or two more mutants.

2) In several of the analyses, the assay conditions had to be modified in order to see differences, e.g. increasing the NaCl concentration to 300mM when assessing complex formation and including phosphorylated eIF2 in the GEF assays. Do the authors have any idea whether such conditions might, or might not, reflect complex formation or GEF activity in vivo? In other words, would the difference in formation or activity in an intact cell be large enough to be physiologically relevant?

3) Subsection “Several VWMD mutations reduce the protein level of the affected subunit in cells”, last paragraph. – What epitope on the α subunit does the anti-eIF2Bα antibody recognize? If it is near V183, could the apparent reduction in α content be due to altered antibody affinity for the protein, rather than an actual decrease in the amount of protein expressed?

4) As noted by the authors, the relative expression of eIF2Bε is increased in some transformed cells compared to non-transformed cells. Do the HEK293T cells used in this study express high levels of eIF2Bε? If so, why were they used for this study rather than a cell line that has lower relative expression?

5) – The authors speculate that "the basal activity of the δR483W mutant is so compromised that ISRIB-stimulated formation of even minimal levels of decamer have a drastic effect on GEF activity". However, the αV183F mutant has similarly low GEF activity (Figure 2) and ISRIB caused a dramatic increase in formation of the decamer. Yet, the magnitude of the difference in GEF activity in the presence and absence of ISRIB is much lower than for the δR483W mutant. Please explain.

[Editors' note: further revisions were requested prior to acceptance, as described below.]

Thank you for submitting your article "The small molecule ISRIB rescues the stability and activity of Vanishing White Matter Disease eIF2B mutant complexes" for consideration by *eLife*. Your article has been evaluated by the Reviewing Editor Alan Hinnebusch and Randy Schekman as the Senior Editor.

The manuscript was judged to be much improved by the addition of new data and revisions of text. There are however a few issues that should be addressed prior to publication, as follows.

- In the statement in the fourth paragraph of the subsection “Recombinant eIF2B with VWMD mutations exhibit impaired GEF activity that is rescued by ISRIB” that the activity of the tetramer is <30% of the decamer and stimulated 2.2X by ISRIB, it is unclear how these values were calculated from the data in Figure 2; and this needs to be made transparent.

- In the second paragraph of the subsection “Recombinant eIF2B with VWMD mutations exhibit reduced decamer stability that is rescued by ISRIB”, they did not cite Figure 2 or Figure 2—figure supplement 2 for the results on any of the mutants in the absence of ISRIB; but should do so.

- Subsection “Recombinant eIF2B with VWMD mutations exhibit reduced decamer stability that is rescued by ISRIB”, last paragraph, first sentence: the word "presumably" should be added, as this is an interpretation, even if a plausible one.

- Subsection “ISRIB attenuates ISR induction in all VWMD cell lines”, third paragraph: It appears that even this statement about levels of ATF4 protein induction "being qualitatively similar" is not justified, as the Western results for ATF4 protein in WT versus mutant extracts are contained on different gels/blots and thus cannot be compared – as they indicated in their rebuttal. They would have to conduct a Western blot of a gel containing extracts from mutants and WT analyzed side by side. Otherwise, remove/modify the statement.

- A loading control, or image of Ponceau S staining, is required to ensure equal loading of extracts for the new Figure 4—figure supplement 4.

---

## [Author Response]

Essential revisions:1) As a follow up to Figure 1, document the extent of holocomplex formation achieved on mixing WT tetramers and α2 dimers.

Done, and included in Figure 1.

2) Provide a rigorous statistical analysis of differences in GEF activity and holocomplex assembly for the different mutant complexes and adjust conclusions accordingly.

Statistical analyses have been performed as requested. Figure panels and text have been updated accordingly. Our conclusions for the activity of each mutant remain unchanged.

3) Conduct experiments to determine whether ISRIB increases the GEF activity of the tetramer by enhancing octamer formation.

We performed SEC runs on the tetramer in the absence and presence of ISRIB to show that this is indeed the case. The new data are presented as Figure 2—figure supplement 1. Furthermore, this finding is consistent with two new cryo-EM structures of eIF2B reported by the groups of Peter Walter (https://www.biorxiv.org/content/early/2017/11/24/222257; doi: 10.1101/222257) and David Ron (https://www.biorxiv.org/content/early/2017/11/25/224824; doi: 10.1101/224824). In particular, the Walter group solved a structure of an ISRIB-stabilized octamer in the absence of eIF2Bα and further validated their finding using analytical ultracentrifugation of the complex, which is orthogonal to our method.

4) Resolve discrepancies between the two panels of Figure 2 regarding GEF activity of the tetramer, clearly shown on the left panel but seemingly contradicted by the right panel.

In the right panel of Figure 2, we background-subtracted the activity of tetramer alone in order to calculate the K_m_ of the eIF2Bα subunits. Thus, it appeared as if the tetramer had zero activity. We have updated the graph title and figure legend to clarify, and moved the panel to Figure 2—figure supplement 1 to make Figure 2 more clear.

5) Improve Figure 2/3A by showing western blots of fractions from the size exclusion chromatography experiments.

We performed the Western blots of the fractions as requested, probing for the δ subunit (representing the 4mer subcomplex) and the α subunit. The data are consistent with the UV chromatograms in Figure 2, albeit less sensitive due to the poorer resolution of elution volume. We have presented the data as Figure 2—figure supplement 2.

6) Quantify the ATF4 protein expression for the experiments shown in Figure 4/Figure 4—figure supplement 2, and discuss any differences that might exist among the mutants.

We previously attempted to quantify basal ATF4 levels, but this was not feasible due to the very low expression of the protein in unstressed HEK293T cells. Across replicate experiments, we have not observed consistently different basal ATF4 levels between WT and the various mutants, as we pointed out in the text. Thus, we chose to restrict ourselves to only qualitative comparisons for the experiment in those panels Note that the panels of WT and mutants should not be compared to each other, but only to treatments within each cell line (hence the separate boxes for each cell line). In fact, the apparent higher ATF4 levels in two of the mutant clones (δR483W #2 and εR113H #2 in Figure 4—figure supplement 2) is due to increased exposure time to account for lower induction of ATF4 upon Tm treatment, for unknown clone-specific reasons.

7) Examine the GEF activity of the recombinant R113H mutant complex, which has been analyzed in all previous studies on VWM mutant complexes.

We measured the GEF activity and holocomplex formation of the recombinant εR113H mutant, and incorporated the data into revised panels in Figure 2. We found that this mutant has activity that is significantly reduced compared to WT and comparable to the εR195H mutant.

8) Measure eIF2B expression levels in HEK cells to determine if they truly are elevated compared to untransformed cells.

We measured eIF2B subunit levels in a panel of five cell lines and discovered that HEK293T cells do indeed have significantly elevated levels, even in comparison to another transformed cell line (HeLa). The data are presented in a new panel (Figure 4—figure supplement 4).

9) Discuss the physiological relevance of using high salt to measure effects of VWM mutations on complex formation; and possibly refrain from inferences that the in vivo defects conferred by V183F or R483W arise from weaker heteropentamer assembly.

We have discussed this point in detail in our response to reviewer 3 below. While the high salt is obviously not reflective of physiological conditions, the highly artificial nature of the SEC assay (high concentration of purified protein, no other macromolecules, non-equilibrium conditions) already make it difficult to compare to conditions in vivo. Ultimately, however, we do believe that the differences revealed in our assay are reflective of physiologically relevant changes caused by the different mutations. Nevertheless, we altered the text in the manuscript slightly to make it clear that we used non-physiological conditions to increase the dynamic range of the assay.

10) Re-think the interpretation of the lower ISRIB EC50 for the R483W variant, which doesn't seem to make sense.

We have included new text in the Discussion to elaborate on this topic, and made clear that our interpretation is a postulation. As such, we do not emphasize this point. However, we believe that our reasoning is sound. We elaborate further on this in the response to reviewer 1 below.

Reviewer #1:General critique:The results are significant in providing direct evidence that heteropentamer formation is critical for robust eIF2B activity in vitro, and underlies the stimulatory effect of the α-α homodimer on GEF activity of the tetramer formed by the other four eIF2B subunits; and that ISRIB rescues GEF activity by promoting heteropentamer formation. The results are also significant in showing that the deleterious effects of 6 different VWM mutations on GEF activity can be rescued to WT levels by ISRIB. The findings that the age of disease onset does not correlate with the severity of the defect in GEF activity for either recombinant complexes or eIF2B complexes in cell lysates represents a rigorous set of results that are germane to this controversial aspect of VWM. Their finding that none of the VWM substitutions generated in HEK cells by gene editing induce the ISR in the absence of stress, nor exacerbate it when PERK is activated, was surprising and might reflect the abnormally high eIF2B levels in the transformed HEK cells. While this is a disappointing aspect of the study, it does not prevent them from showing that ISRIB can rescue eIF2B activity in cell lysates for all 5 VWM mutants tested.1) MALS analysis should be done on the mixture of WT 4-mer and α2 dimers to show the yield of 10-mer achieved in the reconstitutions analyzed in Figure 1.

Done, and included in Figure 1.

2) Subsection “Correlation between eIF2B GEF activity and disease severity of VWMD mutants”, third paragraph, the claim that R483W is more crippling than V183F in assays of both recombinant eIF2B and in cell lysates is not compelling, as there is no demonstration that these differences are statistically significant. It's also unclear whether the activity of R113H in lysates is significantly higher than that of V183F because the variance between the two V183F replicate cell lines is quite large. These interpretations may have to be withdrawn.

We found that the difference in GEF activity between recombinant R483W and V183F is highly statistically significant. However, in cell lysates (when combining both clones of each mutant), the R483W and V183F mutants were not statistically significant (p = 0.08). Therefore, we removed the text stating that R483W was more crippling than V183F in both systems. The difference between R113H and V183F cell lysates is statistically significant, thus that part of our conclusions is unchanged.

3) The statement that "A key finding of our work is that the catalytic activity of recombinant eIF2B is proportional to the stability of the decameric holoenzyme." is an overstatement, as L487W has a smaller effect on decamer stability but a greater effect on activity vs. V183F. What they can say is that several mutations with strong effects on decamer stability also have strong effects on activity, and both defects are co-rescued by ISRIB.

We removed the sentence, as suggested by the reviewer.

Reviewer #2:[…] In general the work appears to have been performed to a high standard, however there are some issues relating to how the experiments are described and presented that should be addressed. In general this study agrees with many prior analyses of VWM alleles in that studies of purified mutant eIF2B can reveal defects, when some cell-based assays do not, and vice versa. The main novel features of this study are the use of CRISPR to make cells lines and the evaluation of ISRIB as a potential curative agent. The conclusions could be strengthened by some additional experiments, which the authors have resources in place to complete.1) Figure 2. Left panel. This figure appears to show that in the absence of eIF2Bα, ISRIB can still stimulate 4-mer GEF activity. Reactions without eIF2Bα are not shown, but were presumably done? The lowest point is 0.01M eIF2Bα, meaning that almost all eIF2B lacks eIF2Bα, but its activity is still boosted by ISRIB. This is not discussed in the manuscript.

The x-axis of the graph was mislabeled due to plotting on a log scale. The lowest point was actually 4-mer in the absence of any a subunit. The error has been corrected. As the reviewer inferred, ISRIB does indeed boost the activity of 4-mer alone (see point below).

How does ISRIB boost 4-mer activity? Does it stabilize octamer formation in the absence of eIF2Bα? Or does ISRIB boost GEF activity via a different mechanism? Performing size exclusion chromatography of the 4mer alone +/- ISRIB would allow evaluation of the first mechanism. It was shown previously that, unlike the mammalian complexes studied here, the yeast eIF2B complex octamer lacking eIF2Bα is stable (see Gordiyenko et al. PMID:24852487).

We performed SEC runs on the tetramer in the absence and presence of ISRIB to show that it stabilizes octamer formation, as the reviewer suspected. The new data are presented as Figure 2—figure supplement 1. Furthermore, this finding is consistent with two new cryo-EM structures of eIF2B reported by the groups of Peter Walter (https://www.biorxiv.org/content/early/2017/11/24/222257; doi: 10.1101/222257) and David Ron (https://www.biorxiv.org/content/early/2017/11/25/224824; doi: 10.1101/224824). In particular, the Walter group solved a structure of an ISRIB-stabilized octamer in the absence of eIF2Bα and further validated their finding using analytical ultracentrifugation of the complex, which is orthogonal to our method. We speculate that the stability of the yeast eIF2B octamer may explain why eIF2Bα is dispensable for viability in that organism – it is not required to “activate” the complex.

2) Figure 2. Similarly there is evidence that both yeast and mammalian eIF2B complexes lacking eIF2Bα are less sensitive to eIF2 phosphorylation, (e.g. Elsby et al. PMID:21795329; Kimball et al. PMID:9582312 for mammalian references). By performing an experiment analogous to the one shown in Figure 1, with 4mer eIF2B +/- ISRIB and V183F +/- ISRIB, the impact of loss of eIF2Bα on sensitivity to the ISR and ISRIB could be assessed. This would enable a comparison of the effects of phospho-eIF2 between purified V183F which has a significant 4mer form (Figure 2) and the 293 cell data in Figure 4.

Our data in Figure 2 show that 4mer responds to ISRIB in the absence of eIF2Bα, and that ISRIB acts by stabilizing an octameric complex. This is entirely consistent with the reported cryo-EM data from the Walter group. Because of the low activity of the tetramer in the absence of eIF2Bα, it is challenging to obtain results in a phospho-eIF2α titration GEF experiment. However, this in and of itself is qualitative evidence that at least in our system, 4mer is sensitive to eIF2 phosphorylation status. This would be consistent with new work from Bogorad et al. (Nucleic Acids Res. 2017) elucidating the binding mechanism of phosphorylated eIF2α to eIF2B – they propose a possible binding surface involving the β and δ subunits of eIF2B.

3) Figure 2 right panel. This presentation is not clear. The plots appear to intersect through 0,0 where the axes meet. This would indicate that the 4mer protein is inactive, which it is not. It be simply that the graph title and legend require changing to indicate what is actually being shown here.

For the purposes of fitting/calculating the K_m_ for WT and V183F eIF2Bα, we needed to generate Michaelis-Menten curves that began at the origin (0,0). Thus, we subtracted the basal rate of 4mer alone (the no α condition). Therefore, the left-most point on each curve does indeed begin at 0,0. We have updated the graph title and figure legend to clarify, and moved the panel to Figure 2—figure supplement 1 to make the main figure cleaner.

4) Figure 2 would be improved by showing example western blots of fractions from the size exclusion chromatography. This should demonstrate that eIF2Bα is absent from the 4mer fraction and may indicate what is present in the low molecular weight fractions. The panels could be presented as a figure supplement.

We performed the Western blots of the fractions as requested, probing for the δ subunit (representing the 4mer subcomplex) and the α subunit. The data are consistent with the UV chromatograms in Figure 2, albeit less sensitive due to the poorer resolution of elution volume. We have presented the data as Figure 2—figure supplement 2.

5) Figure 3 and discussion on in subsection “Correlation between eIF2B GEF activity and disease severity of VWMD mutants”. If the authors have the ability to generate labelled ISRIB, they could determine if its affinity for eIF2B is altered by L487W as they suggest, or is altered for the 4mer vs. 10mer forms.

We attempted to generate ^3^H-labeled ISRIB to perform the requested experiment, but were unsuccessful due to issues with the compound’s poor solubility. However, the cryo-EM structure from David Ron’s group lab specifically highlights δL487 as a residue that forms part of the hydrophobic pocket for ISRIB’s aryl groups. We have also performed loading/release experiments using an unpublished ^3^H-labeled ISRIB analog and confirmed that the δL487W abrogates compound binding (data not shown).

6) Figure 4/Figure 4—figure supplement 2 and text p8 paras 3/4. Figure 4—figure supplement 2 shows that some mutants do have higher ATF4 levels. This is not indicated in the text.

This comment was answered in the “Essential revisions” section above.

7) Figure 4. This figure does not show data for untreated cells. From the data shown in Figure 4—figure supplement 3, it appears the presentation of 4D is over-normalised and therefore does not show clearly where there is any differences in the eIF2B mutant responses to Tg treatment, only responses to ISRIB. By presenting the data differently more information would be clearer.

We replaced Figure 4 with a new panel that includes the data for untreated cells as well as cells treated with ISRIB alone. We have also updated the supplementary panel (now Figure 4—figure supplement 3) to show the EC_50_ values for the response of the cells to Tg treatment. As can be seen, the EC_50_ of Tg treatment is similar between the different lines, indicating similar sensitivity to the stressor. In addition, in Figure 4, we removed the redundant treatment condition of 1 mM DTT to make the panel more clear.

Reviewer #3:1) As noted by the authors, some studies suggest that there might be a correlation between eIF2B GEF activity and age of onset in VWMD (e.g. the Fogli and Horzinski references in the present manuscript) whereas other found no association (e.g. the Liu and van Kollenburg references). Based on these discrepancies the authors logically say that further work is needed to clarify the relationship between GEF activity and VWMD. It is therefore surprising that, other than the δR483W mutant, they did not assess the activity of other mutants assessed in previous studies. For example, the four previous studies that are referenced in this paper all showed that the activity of the εR113H mutant was lower than the wild type protein. In the present study, the assembly of that mutant into the holocomplex was examined, but not its GEF activity. The previous studies showed that other homozygous mutants, e.g. εP243L and βE213G, had lower GEF activity compared to wild type. The authors should confirm that their assay replicates the findings previously reported for at least one or two more mutants.

We purified the recombinant εR113H mutant, and measured its GEF activity and holocomplex formation. Our results are consistent with previous finding, as well as our data from cell lysates with the same mutant, i.e. εR113H has impaired activity relative to WT and comparable activity to the εR195H mutant. We have incorporated this data into revised panels in Figure 2.

2) In several of the analyses, the assay conditions had to be modified in order to see differences, e.g. increasing the NaCl concentration to 300mM when assessing complex formation and including phosphorylated eIF2 in the GEF assays. Do the authors have any idea whether such conditions might, or might not, reflect complex formation or GEF activity in vivo? In other words, would the difference in formation or activity in an intact cell be large enough to be physiologically relevant?

The reviewer makes a good point. Our cursory estimations of the amount of eIF2B subunits in cells/tissues suggests that there is significant variation between cell types (partly illustrated in Figure 4—figure supplement 4), ranging from single-digit nM to the low hundreds of nM. In cells, macromolecular crowding and potential binding partners may also influence complex stability. Our SEC experiments are performed with μM concentrations of purified protein. Presumably, instead of raising salt concentration, we could have achieved the same effect by lowering protein concentration, but this would place it below the detection limits of the assay. Ultimately, we believe that the differences revealed by our high salt SEC assay are reflective of real differences that occur in vivo. Concentration differences in different cell types could explain why somatic eIF2B mutations only manifest as pathology in the brain – perhaps levels of the subunits are particularly low in cells of that tissue. The use of phospho-eIF2 in our assays was not required to observe differences between the various mutants, but only to increase the dynamic range of the assay to ISRIB, in order to obtain better EC_50_ measurements. However, the inclusion of a small amount of phospho-eIF2 is likely a good mimic of the situation in vivo, where low level phosphorylation is present. It is particularly notable that progression of VWMD is exacerbated by insults such as head trauma or viral infection, which would trigger eIF2 phosphorylation.

3) Subsection “Several VWMD mutations reduce the protein level of the affected subunit in cells”, last paragraph. What epitope on the α subunit does the anti-eIF2Bα antibody recognize? If it is near V183, could the apparent reduction in α content be due to altered antibody affinity for the protein, rather than an actual decrease in the amount of protein expressed?

We used a polyclonal antibody raised against the entire 305 aa sequence of eIF2Bα. Thus, we believe it is unlikely that the observed reduction in α content is simply due to a difference in epitope recognition.

4) As noted by the authors, the relative expression of eIF2Bε is increased in some transformed cells compared to non-transformed cells. Do the HEK293T cells used in this study express high levels of eIF2Bε? If so, why were they used for this study rather than a cell line that has lower relative expression?

We measured eIF2B subunit levels in a panel of five cell lines and discovered that HEK293T cells do indeed have significantly elevated levels (including eIF2Bε) when normalized to total protein, even in comparison to another transformed cell line (HeLa). The data are presented in a new panel (Figure 4—figure supplement 4). The use of HEK293Ts was due to the existence of the previously-characterized ATF4 reporter cell line (used in Sidrauski et al., 2013) as well as their ease of transfection, which facilitated generation of the VWMD mutants by CRISPR.

5) The authors speculate that "the basal activity of the δR483W mutant is so compromised that ISRIB-stimulated formation of even minimal levels of decamer have a drastic effect on GEF activity". However, the αV183F mutant has similarly low GEF activity (Figure 2) and ISRIB caused a dramatic increase in formation of the decamer. Yet, the magnitude of the difference in GEF activity in the presence and absence of ISRIB is much lower than for the δR483W mutant. Please explain.

This point was partially addressed as part of our responses to the Essential revisions and reviewer 1 above. We stress the difference between the concentration regimes used in the GEF assay (nM) and the SEC assay (μM). ISRIB’s ability to promote decamer function is limited if the α subunit itself has reduced affinity for the 4mer subcomplex (as is the case for αV183F). This is manifested as partial rescue of GEF activity when concentrations of the eIF2B subunits are low in the assay. In the SEC assay, concentrations of the eIF2B subunits are high and thus a large rescue of complex stability is observed. In Figure 2, we show that using μM concentrations of αV183F while maintaining nM concentrations of 4mer, we can achieve complete rescue of GEF activity.

[Editors' note: further revisions were requested prior to acceptance, as described below.]

The manuscript was judged to be much improved by the addition of new data and revisions of text. There are however a few issues that should be addressed prior to publication, as follows.- In the statement in the fourth paragraph of the subsection “Recombinant eIF2B with VWMD mutations exhibit impaired GEF activity that is rescued by ISRIB” that the activity of the tetramer is <30% of the decamer and stimulated 2.2X by ISRIB, it is unclear how these values were calculated from the data in Figure 2; and this needs to be made transparent.

We added text to the paragraph to explain which data points were used to calculate the values. In the process, we updated the value of ISRIB stimulation slightly to be more accurate (from 2.2X to 2.3X).

- In the second paragraph of the subsection “Recombinant eIF2B with VWMD mutations exhibit reduced decamer stability that is rescued by ISRIB”, they did not cite Figure 2 or Figure 2—figure supplement 2 for the results on any of the mutants in the absence of ISRIB; but should do so.

Done.

- Subsection “Recombinant eIF2B with VWMD mutations exhibit reduced decamer stability that is rescued by ISRIB”, last paragraph, first sentence: the word "presumably" should be added, as this is an interpretation, even if a plausible one.

Added.

- Subsection “ISRIB attenuates ISR induction in all VWMD cell lines”, third paragraph: It appears that even this statement about levels of ATF4 protein induction "being qualitatively similar" is not justified, as the Western results for ATF4 protein in WT versus mutant extracts are contained on different gels/blots and thus cannot be compared – as they indicated in their rebuttal. They would have to conduct a Western blot of a gel containing extracts from mutants and WT analyzed side by side. Otherwise, remove/modify the statement.

We removed the statement about qualitative similarity. Our reporter experiment in Figure 4 is a better way of comparing across wild-type and mutant cells, with the added advantage of being a quantitative assay.

- A loading control, or image of Ponceau S staining, is required to ensure equal loading of extracts for the new Figure 4—figure supplement 4.

We avoided using a common “loading control” (e.g. GAPDH, tubulin or actin) that would likely differ in concentration between different cell lines. We instead show the direct equivalent of a Ponceau stain, i.e. the total protein loaded into the capillaries of the Wes instrument. This confirms equal loading of total protein across our samples. We have updated Figure 4—figure supplement 4 with the new data.